# META-PROMPT OPTIMIZATION FOR LLM-BASED SEQUENTIAL DECISION MAKING

## ABSTRACT

Large language models (LLMs) have recently been employed as agents to solve sequential decision-making tasks such as Bayesian optimization and multi-armed bandits. These works usually adopt an LLM for sequential action selection by providing it with a fixed, manually designed *meta-prompt*. However, numerous previous works have found that the prompt has a significant impact on the performance of the LLM, which calls for a method to automatically optimize the meta-prompt for LLM-based agents. Unfortunately, the non-stationarity in the reward observations during LLM-based sequential decision making makes meta-prompt optimization highly challenging. To address this challenge, we draw inspirations from *adversarial bandit* algorithms, which are inherently capable of handling non-stationary reward observations. Building on this foundation, we propose our *EXPonential-weight algorithm for prompt Optimization* (EXPO) to automatically optimize the task description and meta-instruction in the meta-prompt for LLM-based agents. We also extend EXPO to additionally optimize the exemplars (i.e., history of interactions) in the meta-prompt to further enhance the performance, hence introducing our EXPO-ES algorithm. We use extensive experiments to show that our algorithms significantly improve the performance of LLM-based sequential decision making.

## 1 INTRODUCTION

The strong capabilities of LLMs have spurred significant recent interests in adopting them as agents to solve sequential decision-making problems, such as multi-armed bandits (MAB) (Krishnamurthy et al., 2024), Bayesian optimization (BO) (Yang et al., 2024) and reinforcement learning (RL) (Dai et al., 2024). Specifically, these methods often use an LLM to sequentially select the actions by providing it with a specially designed prompt, which we refer to as the *meta-prompt*. The meta-prompt often contains several components, such as the *task description*, the *meta-instruction* (which is used to instruct the LLM to select an action in every step), the *history of interactions* with the environment, among others. The previous methods have all adopted a fixed, manually designed meta-prompt for the LLM-based agent throughout the entire sequential decision-making process. However, numerous previous works have highlighted that the output text generated by LLMs is heavily dependent on its input prompt (Zhou et al., 2023). Therefore, using fixed, manually designed meta-prompt may significantly limit the performance of the LLM-based agents, because handcrafted prompts are often far from optimal (Zhou et al., 2025). This naturally begs the question: *can we automatically optimize the meta-prompt for LLM-based agents to enhance their performance?*

The sensitivity of LLM-generated text to its input prompt has given rise to many recent works on *automated prompt optimization*, among which a representative line of works have adopted the method of multi-armed bandits (MAB) to automatically optimize the prompt (Lin et al., 2024a;b; Wu et al., 2024). Unfortunately, the problem of meta-prompt optimization for LLM-based agents presents significant challenges compared to traditional prompt optimization. This is mostly due to the *non-stationarity in the observed rewards* during the LLM-based sequential decision-making process. Specifically, as the LLM-based agent engages in more interactions with the environment, its state in the environment changes, making its observed rewards non-stationary. For example, in MAB (Krishnamurthy et al., 2024) and BO (Yang et al., 2024), the observed rewards in later iterations (i.e., after the agent has accumulated significant experience in the environment) tend to be higher than those obtained in initial iterations. Similarly, in RL (Dai et al., 2024), rewards are typically dependent on

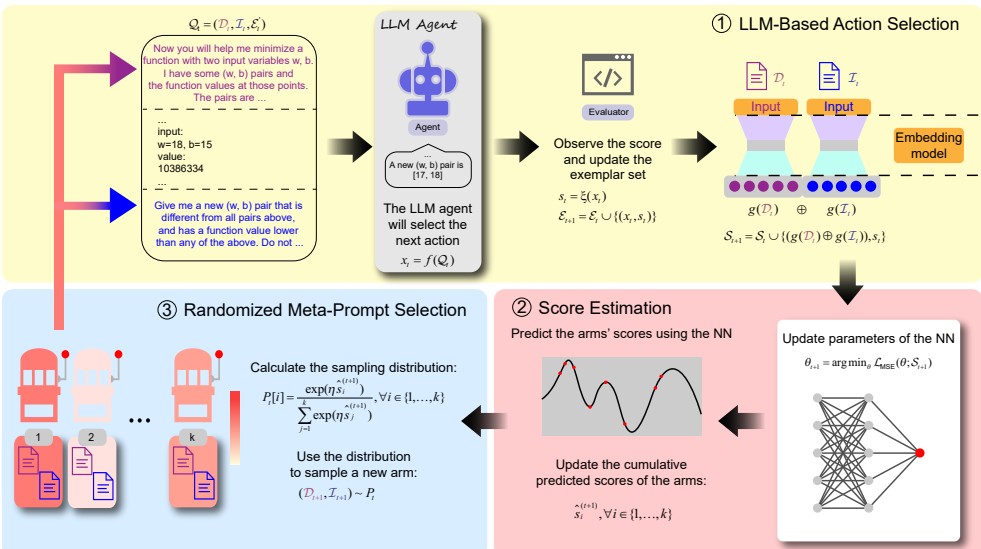

Figure 1: Illustration of our EXPO algorithm. We use purple to denote the task description and blue to represent the meta-instruction.

both the state and action. However, since the state of the LLM-based agent evolves across iterations, this also results in non-stationarity in the observed rewards. As a consequence of the non-stationarity, for the same meta-prompt (e.g., the same task description and meta-instruction), its corresponding observed reward is highly likely to be dynamically changing across different iterations. This is in stark contrast to classical prompt optimization, in which the reward or score for a prompt remains stationary across iterations. As a result, this renders the previous works on prompt optimization (such as those based on MAB (Lin et al., 2024a;b; Wu et al., 2024)) inapplicable, and hence calls for novel algorithmic designs to solve the problem of meta-prompt optimization for LLM-based agents. To this end, we draw inspirations from the field of *adversarial bandits* (Lattimore & Szepesvári, 2020).

In adversarial bandits, for each arm, the reward observations when the arm is pulled are chosen by an adversary, i.e., they are allowed to change in an arbitrary way across different iterations. Therefore, the reward observations can be significantly non-stationary. This is considerably different from classical stochastic MAB, in which the reward observations for an arm are sampled from a fixed stationary distribution. Therefore, the ability of adversarial bandits to handle non-stationary reward observations makes it an ideal candidate for meta-prompt optimization for LLM-based agents. Specifically, drawing inspirations from the EXP3 algorithm for adversarial bandits, we introduce our *EXPonential-weight algorithm for prompt Optimization* (EXPO) to optimize the task description and meta-instruction in the meta-prompt of an LLM-based agent.

In addition to the task description and meta-instruction, the *history of interactions* with the environment (which we also refer to as the *exemplars*) is also a crucial component in the meta-prompt which exerts a considerable impact on the performance of LLM-based agents. Existing works often adopt simple heuristic approaches to decide how to incorporate the exemplars into the meta-prompt, including which subset of exemplars is included and their ordering in the meta-prompt. Previous works on in-context learning (ICL) have found that in addition to their contents, the ordering of the exemplars also has a significant impact on the performance of LLMs (Lu et al., 2022). Therefore, in addition to optimizing the task description and meta-instruction, we also extend our EXPO algorithm to additionally optimize both the subset of exemplars included in the meta-prompt and their ordering. However, the optimization of the task description and meta-instruction in every iteration in our EXPO makes the optimization of exemplars non-stationary as well. Specifically, for the same subset of exemplars with a fixed ordering, their reward observations are usually non-stationary, because the task description and meta-instruction selected by our EXPO algorithm are highly likely to vary across different iterations. To this end, we extend our EXPO algorithm to additionally use a separate adversarial bandit method to optimize the exemplars (i.e., the interaction history) in the meta-prompt for LLM-based agents, and hence introduce our *EXPO with Exemplar Selection* (EXPO-ES) algorithm.

We use extensive experiments to show that our EXPO significantly improves the performance of the LLM-based BO algorithm from Yang et al. (2024) (Sec. 4.1) and the LLM-based MAB algorithm

from Krishnamurthy et al. (2024) (Sec. 4.2). Furthermore, in tasks where the exemplars provide crucial information for the LLM-based agent, our `EXPO-ES` further enhances the performance of `EXPO` via automated exemplar selection (Sec. 4.1). We also use ablation study to unveil interesting insights about our algorithms (Sec. 5).

## 2 PROBLEM SETTING

Throughout our work, we use *arms* to represent meta-prompts, and use *actions* to denote the actions selected by an LLM-based agent. Consider an algorithm which uses an LLM to perform a sequential decision-making task by sequentially instructing the LLM to select an action in every iteration. A representative example of such algorithms is the *Optimization by PROmpting* (OPRO) algorithm from Yang et al. (2024). OPRO aims to solve an optimization problem, i.e., to find $x^* = \arg\min_x f(x)$. To achieve this, in every iteration $t$, OPRO uses an LLM to select a batch of $B$ input queries $\{x_{t,1}, \ldots, x_{t,B}\}$, after which their corresponding scores $\{s_{t,1}, \ldots, s_{t,B}\}$ are observed. When instructing the LLM to select the input queries, the meta-prompt $\mathcal{Q}$ given to the LLM contains a number of important components, including *a fixed task description* $\mathcal{D}$, *a fixed meta-instruction* $\mathcal{I}$, and *a sequence of exemplars* $\mathcal{E}_t$ corresponding to a subset of the observations (i.e., pairs of input queries and observed scores) collected so far. The same paradigm of LLM-based sequential decision making has also been adopted by other works, such as the LLM-based MAB algorithm from Krishnamurthy et al. (2024) (more details in Sec. 4.2).

In this work, our first algorithm, `EXPO` (Sec. 3.1), dynamically optimize the task description $\mathcal{D}$ and meta-instruction $\mathcal{I}$ (i.e., selects a new $\mathcal{D}_t$ and $\mathcal{I}_t$ in every iteration $t$), in order to improve the efficiency and effectiveness of optimization. We also extend our `EXPO` to derive the `EXPO-ES` algorithm (Sec. 3.2), which additionally optimizes the sequence of exemplars $\mathcal{E}_t$ to further improve the optimization performance. We use $g(\cdot)$ to denote a pre-trained embedding function, which maps some input text to its corresponding continuous representation. We separately obtain the embeddings of the task description $g(\mathcal{D}_t)$, the meta-instruction $g(\mathcal{I}_t)$ and the exemplar sequence $g(\mathcal{E}_t)$. Based on the embeddings, in every iteration, we use the current history of selected meta-prompts and their scores to train a neural network (NN), which can then be used to predict the scores of every meta-prompts in the domain. We denote this NN as $\mathcal{M}(g(\cdot); \theta)$, in which $\theta$ represents the NN parameters.

**Adversarial Bandits.** In adversarial bandits, the goal is to compete against the best arm *in hindsight* (Lattimore & Szepesvári, 2020). Consider an MAB problem with $k$ arms (i.e., meta-prompts). For each arm $i = 1, \ldots, k$, denote its corresponding sequence of rewards (i.e., scores) in $T$ iterations as $\{r_{t,i}\}_{t=1,\ldots,T}$. The best arm in hindsight is then defined as $i^* = \arg\max_{i=1,\ldots,k} \sum_{t=1}^{T} r_{t,i}$. Then, the goal of an adversarial bandit algorithm (which selects arm $A_t$ in iteration $t$) is to minimize the following definition of regret: $R_T = \sum_{t=1}^{T} r_{t,i^*} - \sum_{t=1}^{T} r_{t,A_t}$.

**Adversarial Bandits for LLM-Based Agents.** LLM-based sequential decision-making methods often aim to maximize either *(a)* the cumulative rewards (e.g., the LLM-based MAB algorithm from Krishnamurthy et al. (2024)) or *(b)* the final reward (e.g., OPRO from Yang et al. (2024)). In the former case of cumulative reward maximization, the overall rewards/scores for the best arm $i^*$ are higher than the other arms. In the latter case, we implicitly assume that the arm with the largest final reward after $T$ iterations also has large rewards across all iterations in general. As a result, in both cases, *the observed rewards of an arm (i.e., the observed scores of a meta-prompt) in every iteration are indicative of the quality of the arm (i.e., the meta-prompt)*. So, when training the NN $\mathcal{M}(g(\cdot); \theta)$ (for score prediction) using the history of the selected meta-prompts and their observed scores, we simply use the scores (i.e., rewards) as the labels in the training set. This simple design helps our algorithms achieve strong performance in our experiments (Sec. 4).

## 3 ALGORITHMS

### 3.1 THE `EXPO` ALGORITHM (ALGO. 1)

**Domain Generation.** At the beginning of our algorithm, we start by generating the domain of task descriptions and meta-instructions. Following the previous works on prompt optimization (Lin et al., 2024a;b; Zhou et al., 2023), we use an LLM to rephrase an initial task description $\mathcal{D}_0$ (resp. initial meta-instruction $\mathcal{I}_0$) to generate a domain of $k_1$ task descriptions (resp. $k_2$ meta-instructions). This results in a domain size of $k = k_1 \times k_2$. We defer more details on domain generation to App. D.1. We treat the combination of a task description $\mathcal{D}$ and a meta-instruction $\mathcal{I}$ in the domain as an *arm*,

i.e., our adversarial bandit problem has $k$ arms. In addition to jointly optimizing $\mathcal{D}$ and $\mathcal{I}$, we have also evaluated the performance of optimizing them separately. The results (Fig. 6) show that jointly optimizing these two components leads to better performance.

① **LLM-Based Action Selection (lines 3-7 of Algo. 1).** At the beginning of every iteration $t$, we firstly use the current task description $\mathcal{D}_t$, meta-instruction $\mathcal{I}_t$ and exemplar sequence $\mathcal{E}'_t$ selected at the end of the last iteration $t-1$ (more details below) to construct a meta-prompt $\mathcal{Q}_t = (\mathcal{D}_t, \mathcal{I}_t, \mathcal{E}'_t)$ (line 3). Then, we use $\mathcal{Q}_t$ as the input prompt to the LLM $f(\cdot)$ to select the next action $x_t$ and collect its score $s_t$ (lines 4-5). After that, we update the set of exemplars $\mathcal{E}_t$ and the meta-prompt-score set $\mathcal{S}_t$ (lines 6-7).

② **Score Estimation (lines 8-9).** In the classical EXP3 algorithm for adversarial bandits with a finite number of arms, the cumulative sum of *the observed rewards* of every arm is used to construct the arm sampling distribution through an exponential-weight mechanism (Lattimore & Szepesvári, 2020). However, in problems where the number of arms is excessively large (e.g., our problem of meta-prompt optimization), the reward observations for many arms are not available. Therefore, the cumulative sum of *the estimated rewards* of every arm is often used instead to construct the sampling distribution (Lattimore & Szepesvári, 2020). Therefore, we firstly estimate the scores of all $k$ arms (i.e., meta-prompts) in the domain and then use these score estimates to derive an arm sampling distribution for our EXPO. A number of recent works have shown that using a neural network (NN) (which takes the pre-trained embedding $g(\cdot)$ as input) for score/reward estimation leads to powerful prompt optimization algorithms (Lin et al., 2024a;b; Wu et al., 2024). Therefore, we also adopt an NN $\mathcal{M}(g(\cdot); \theta)$ for score estimation in our EXPO. Specifically, in every iteration $t$, we use the history of selected meta-prompts and their scores, denoted as $\mathcal{S}_{t+1}$ (line 7 of Algo. 1), to train an NN by minimizing the mean-squared error (MSE) loss (line 8 of Algo. 1). The trained NN with parameters $\theta_{t+1}$ can then be used to estimate the score of every arm (i.e., every combination of task description and meta-instruction) in the domain. For every arm, its estimated score is then added to its corresponding *cumulative sum of score estimates* $\hat{s}_i^{(t+1)}$ (line 9 of Algo. 1). Note that every term in the cumulative sum $\hat{s}_i^{(t+1)}$ represents our score estimate for arm $i$ *in a particular iteration* $t$, i.e., our estimated score for arm $i$ from an NN trained using the observation history *up to iteration* $t$. The updated cumulative sums of score estimates $\hat{s}_i^{(t+1)}$ for all $k$ arms are then used for randomized arm (i.e., meta-prompt) selection, which we discuss next.

③ **Randomized Meta-Prompt Selection (lines 10-12).** After the cumulative sum $\hat{s}_i^{(t+1)}$ of every arm $i$ is updated, we follow the EXP3 algorithm (Lattimore & Szepesvári, 2020) and use the cumulative sums to construct a distribution following Equation equation 1. Then, we use this distribution to randomly sample the next arm, i.e., the next task description $\mathcal{D}_{t+1}$ and meta-instruction $\mathcal{I}_{t+1}$ (line 11 of Algo. 1). *Randomization* is a key principle in adversarial bandits (Lattimore & Szepesvári, 2020), and the randomization involved in our arm selection strategy is crucial for the ability of our EXPO to deal with non-stationary reward observations. The heuristic to select a sequence of exemplars $\mathcal{E}'_{t+1}$ (line 12 of Algo. 1) is often specified by the LLM-based sequential decision-making algorithm (Yang et al., 2024). We discuss more details on this, as well as the extension of our EXPO algorithm to automatically select $\mathcal{E}'_{t+1}$, in Sec. 3.2.

**Exploitation vs. Exploration.** Our EXPO algorithm is able to achieve a principled balance between exploitation and exploration. The use of powerful pre-trained embedding and NNs allows us to achieve accurate score estimation. Therefore, the cumulative score estimate $\hat{s}_i^{(t+1)}$ (line 9 of Algo. 1) provides a reliable assessment of the quality of every arm $i$ (i.e., every combination of task description and meta-instruction). This ensures that an arm with a large score is given a large weight in the sampling distribution $P_t$ (line 10) and hence leads to reliable *exploitation*. Meanwhile, the inherent randomness in our randomized arm selection strategy ensures that enough *exploration* is performed in the domain of meta-prompts.

**Batch Action Selection.** In the description of our EXPO (Algo. 1), although we select one action $x_t$ in every iteration $t$, this can be easily generalized to select a batch of actions. For example, when applying our EXPO to improve OPRO (Yang et al., 2024) (Sec. 4.1), we follow the practice of OPRO to select a batch of 8 actions/queries in every iteration (i.e., step 4 of Algo. 1) and set the temperature of the LLM to 1 to ensure the diversity of the selected actions. In order to obtain a noiseless and reliable score to assess the quality of the meta-prompt $\mathcal{Q}_t$, we set the temperature to 0 when selecting the last action and use its corresponding observed score as the score $s_t$ of $\mathcal{Q}_t$ (line 5 of Algo. 1).

---

**Algorithm 1** EXPO

---

**input** Initial task description $\mathcal{D}_0$, initial meta-instruction $\mathcal{I}_0$.

1: Initialize the exemplar set $\mathcal{E}_0 = \emptyset$, and the subset $\mathcal{E}'_0 = \emptyset$, meta-prompt-score set $\mathcal{S}_0 = \emptyset$, and cumulative score estimates $\hat{s}_i^{(0)} = 0$ for all $i \in \{1, \ldots, k\}$.

2: **for** iteration $t = 0, 1, \ldots, T-1$ **do**

3:    Construct meta-prompt $\mathcal{Q}_t = (\mathcal{D}_t, \mathcal{I}_t, \mathcal{E}'_t)$.

4:    Query the LLM $f(\cdot)$ using the meta-prompt $\mathcal{Q}_t$ to select the next action $x_t$: $x_t = f(\mathcal{Q}_t)$.

5:    Observe the score $s_t$ for $x_t$ using the task-specific evaluator: $s_t = \xi(x_t)$.

6:    Update the exemplar set $\mathcal{E}_{t+1} = \mathcal{E}_t \cup \{(x_t, s_t)\}$.

7:    Update the meta-prompt-score set $\mathcal{S}_{t+1} = \mathcal{S}_t \cup \{([g(\mathcal{D}_t) \oplus g(\mathcal{I}_t)], s_t)\}$, where $g(\cdot)$ denotes the embedding function and $\oplus$ denotes concatenation.

8:    Update the parameters $\theta$ of the neural network (NN) $\mathcal{M}(g(\cdot); \theta)$ by using the updated $\mathcal{S}_{t+1}$ as the training set to minimize the MSE loss, yielding $\theta_{t+1}$.

9:    Update the cumulative score estimates $\hat{s}_i^{(t)}$ for all arms $i$ using the predicted scores from $\mathcal{M}(g(\cdot); \theta_{t+1})$: $\hat{s}_i^{(t+1)} = \hat{s}_i^{(t)} + \mathcal{M}([g(\mathcal{D}_i) \oplus g(\mathcal{I}_i)]; \theta_{t+1}), \quad \forall i \in \{1, \ldots, k\}$.

10:    Compute the sampling distribution $P_t$ over all arms:

$$P_t[i] = \frac{\exp(\eta \hat{s}_i^{(t+1)})}{\sum_{j=1}^k \exp(\eta \hat{s}_j^{(t+1)})}, \quad \forall i \in \{1, \ldots, k\} \tag{1}$$

11:    Sample an arm (i.e., the combination of a task description and a meta-instruction) from $P_t$: $(\mathcal{D}_{t+1}, \mathcal{I}_{t+1}) \sim P_t$.

12:    Select a sequence of exemplars $\mathcal{E}'_{t+1}$ from $\mathcal{E}_{t+1}$ following a pre-defined heuristic method.

---

### 3.2 EXPO WITH EXEMPLAR SELECTION (EXPO-ES)

Previous works on LLM-based for sequential decision making often select the sequence of exemplars $\mathcal{E}'_{t+1}$ included in the meta-prompt $\mathcal{Q}_t$ using a fixed pre-defined heuristic (line 12 of Algo. 1). For example, OPRO includes the 20 exemplars with the best observed performance in the meta-prompt, arranged in ascending order of performance (Yang et al., 2024); the LLM-based MAB method from Krishnamurthy et al. (2024) either includes all exemplars (ordered by their iteration sequence) in the prompt or includes a summarized representation of all exemplars. However, numerous previous works have reported that both the subset of exemplars and their ordering have significant impacts on the performance of LLM (Wu et al., 2024). Therefore, here we further extend our EXPO (Algo. 1) to additionally optimize the sequence of exemplars $\mathcal{E}'_{t+1}$ (i.e., to replace line 12 of Algo. 1 by an automated method to select $\mathcal{E}'_{t+1}$), hence introducing our EXPO-ES algorithm (Algo. 2, App. C).

As a result of the dynamically changing task description and meta-instruction, the optimization of exemplar sequences becomes non-stationary as well. So, we also dynamically optimize the exemplar sequence based on the EXP3 algorithm for adversarial bandits. That is, in every iteration of our EXPO-ES (Algo. 2), we firstly optimize the task description and meta-instruction (i.e., following lines 3-11 of Algo. 1), and then optimize the exemplar sequence $\mathcal{E}'_{t+1}$ in a similar way to Algo. 1.

**Details of EXPO-ES (Algo. 2).** Specifically, after the task description and meta-instruction are optimized (i.e., after lines 3-11 of Algo. 1), we firstly extract the embedding of the exemplar sequence $\mathcal{E}'_t$ used in this iteration: $g(\mathcal{E}'_t)$, and add $(g(\mathcal{E}'_t), s_t)$ to the *exemplar training set* $\mathcal{T}_{t+1}$ (line 4 of Algo. 2). Next, the updated dataset $\mathcal{T}_{t+1}$ is used to train an NN with parameters $\theta_{t+1}^{\text{ES}}$ (line 5), which is able to *estimate the score of any exemplar sequence*. Subsequently, we randomly sample $k^{\text{ES}}$ exemplars sequences, each containing $\mathcal{L}$ exemplars, to be used as our *domain of exemplar sequences* (line 8). Next, for every candidate exemplar sequence in the domain, we need to obtain its cumulative score estimate (similar to line 9 of Algo. 1). Unfortunately, due to *the time-varying nature of the domain of exemplar sequences* (due to the addition of new exemplars and random sampling of exemplar sequences), we are no longer able to constantly maintain a cumulative score estimate for every exemplar sequence (i.e., arm) and update incrementally. To this end, we save the parameters of the trained NN in every iteration in history; then for each sampled exemplar sequence in the domain, we obtain its score estimates from all NNs in the history and use their sum as the *cumulative score estimate* for this exemplar sequence (lines 9-14 of Algo. 2). Next, the cumulative score estimates for

---

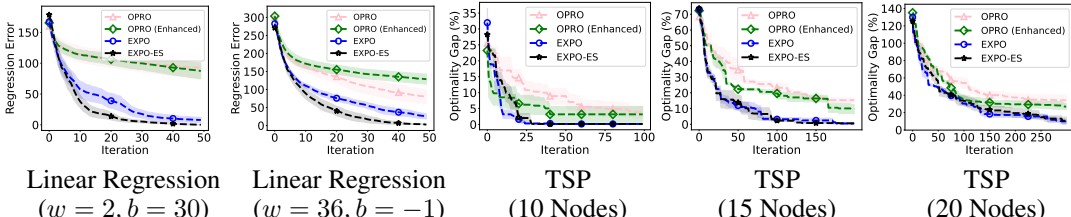

|                          |                          |               |               |               |
| ------------------------ | ------------------------ | ------------- | ------------- | ------------- |
| Linear Regression        | Linear Regression        | TSP           | TSP           | TSP           |
| ($w = 2, b = 30$)        | ($w = 36, b = -1$)       | (10 Nodes)    | (15 Nodes)    | (20 Nodes)    |

Figure 2: Results of different algorithms (mean $\pm$ standard error) in the Linear Regression and TSP task (Sec. 4.1). Lower is better.

all exemplar sequences are used to compute the sampling distribution, from which the next exemplar sequence $\mathcal{E}'_{t+1}$ is sampled and used to the meta-prompt in the next iteration (lines 15-16).

## 4 EXPERIMENTS

We firstly apply our algorithms to improve the performance of OPRO in Linear Regression (LR), traveling salesman problem (TSP) and instruction optimization tasks, adopting the same experimental setting as Yang et al. (2024) (Sec. 4.1). Then, we use our algorithms to enhance the performance of the LLM-based MAB algorithm from Krishnamurthy et al. (2024).

### 4.1 LR, TSP AND INSTRUCTION OPTIMIZATION TASKS

**Linear Regression (LR).** In the LR task, our goal is to find the optimal LR coefficients, $w$ and $b$, that best fit a set of given noisy observations. We firstly choose the groundtruth LR coefficients $w_{\text{true}}$ and $b_{\text{true}}$, and use them to generate noisy observations for 50 inputs $x$ which are randomly selected within $[-1, 1]$. Specifically, for each input $x$, we generate its noisy observation as $y = w_{\text{true}}x + b_{\text{true}} + \epsilon$ where $\epsilon$ is a Gaussian noise. We adopt the two most challenging choices of coefficients from Yang et al. (2024): (1) $w_{\text{true}} = 2, b_{\text{true}} = 30$ and (2) $w_{\text{true}} = 36, b_{\text{true}} = -1$. In this task, OPRO aims to find the optimal $w$ and $b$ which minimizes the regression error (i.e., mean squared error).

**Traveling Salesman Problem (TSP).** In the classical TSP problem (Jünger et al., 1995), given a set of $n$ nodes with their coordinates, the objective is to find the shortest route that starts from a given node, traverses all nodes exactly once, and finally returns to the starting node. Therefore, our goal is to solve a discrete optimization problem in which the input variable is a trajectory and the goal is to minimize the total distance of the trajectory. We adopt TSP instances with 10, 15, and 20 randomly generated nodes, respectively, which represent increasing levels of difficulty.

We adopt GPT-3.5-Turbo as the LLM in both the LR and TSP tasks. The results for both tasks are shown in Fig. 2, which plot the regression error (i.e., mean squared error) for the LR tasks and optimality gap (i.e., the difference between the total distance of the discovered route and that of the optimal route) for the TSP tasks (lower is better for both tasks). Of note, in addition to the standard OPRO (pink curves) (Yang et al., 2024), we have also proposed an enhanced variant of OPRO (green curves) in which we added some further clarifications to the task description (see App. D.2.5 for more details). The enhanced variant consistently improves the performance of the standard OPRO (Fig. 2).[1] More importantly, the results in Fig. 2 show that ***in all tasks, our EXPO algorithm (blue curves) significantly and consistently outperforms OPRO***, including both standard OPRO and its enhanced variant. This demonstrates that our meta-prompt optimization approach, grounded in adversarial bandits, leads to more efficient (i.e., faster convergence) and more effective (i.e., improved final performance) LLM-based sequential decision making.

Meanwhile, our EXPO-ES algorithm, which is additionally equipped with automated exemplar selection, considerably improves the performance of EXPO in the LR tasks yet performs on par with EXPO in the TSP tasks. This is likely because the exemplars play a more important role in the LR tasks than the TSP tasks. Specifically, in LR, *the input-output exemplars provide important information for identifying the optimal LR coefficients* (Wu et al., 2024). Therefore, selecting better exemplars (via our EXPO-ES) brings significant performance boost. On the other hand, in the TSP tasks, due to the challenging nature of the tasks, it is difficult for the LLM to infer crucial and useful information from the exemplars. Therefore, the other components in the meta-prompt (i.e., the task

---

[1]As discussed in the last paragraph of Sec. 3.1, we have slightly modified OPRO to select the last action in the batch using a temperature of 0. We empirically show that this leads to comparable performance with the original OPRO which uses a temperature of 1 to choose all 8 actions (see Fig. 15 in App. E.2).

| OPRO | EXPO |
|---|---|
| Now you will help me minimize a function with two input variables $w, b$. I have some $(w, b)$ pairs and the function values at those points. The pairs are arranged in descending order based on their function values, where lower values are better.

{EXEMPLARS}

Give me a new $(w, b)$ pair that is different from all pairs above, and has a function value lower than any of the above. Do not write code. The output must end with a pair $[w, b]$, where $w$ and $b$ are numerical values. | We will collaborate to optimize a function involving two parameters, $w$ and $b$. I possess a set of data points, each consisting of $(w, b)$ pairs and their corresponding function values. These pairs are systematically organized in reverse order, starting from the greatest to the smallest function values. Essentially, the lower the function value, the more optimal or preferable the pair. Consequently, our goal is to identify and analyze the $(w, b)$ pair that manifests the lowest function value, as this represents the perspective of optimum efficacy.

{EXEMPLARS}

To enhance the quality and expand on the existing instructions, follow these improved guidelines vis-à-vis designing a new and distinctive numerical pair: ensure the selected $(w, b)$ combination diverges from prior examples and secures a function output lower than preceding values. Key details on methodology or calculations are not required—just ensure clarity in presenting a returned value that closes with the specific format $[w, b]$, where both $w$ and $b$ are distinct numerical figures. |

Figure 3: The task description and meta-instruction used by OPRO (left) and optimized by our EXPO (right) in a Linear Regression task.

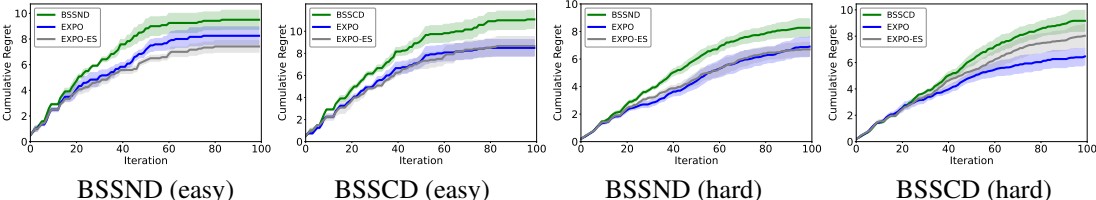

|  |  |  |  |
|---|---|---|---|
| BSSND (easy) | BSSCD (easy) | BSSND (hard) | BSSCD (hard) |

Figure 4: Cumulative regret in the LLM-based MAB experiments (Sec. 4.2). Lower is better.

description and meta-instruction) provide more useful information in the TSP tasks. As a result, *selecting better exemplars does not lead to noticeable performance gains in the TSP tasks*. Fig. 3 provides an illustration of the original task description and meta-instruction used by OPRO and those discovered by our EXPO algorithm for the LR tasks, whereas the corresponding meta-prompts for the TSP tasks are displayed in Fig. 17 in App. E.5.

**Instruction Optimization.** We also adopt our EXPO to improve the performance of OPRO in instruction optimization tasks using the GSM8K dataset (Cobbe et al., 2021). We adopt GPT-4-Turbo as the optimizer LLM to sequentially propose different instructions, and use Qwen-2.5-7B as the scorer LLM to evaluate the effectiveness of these instructions (more details in App. D.2.1). The results in Fig. 5 show that our EXPO also considerably outperforms OPRO in this challenging task. Moreover, EXPO-ES further improves the performance of EXPO, which is likely because the exemplars (i.e., instructions and their scores) provide valuable information for instruction optimization in this task.

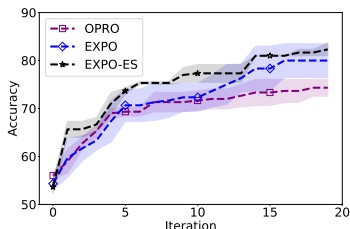

Figure 5: Instruction optimization task (higher is better).

## 4.2 LLM-BASED MULTI-ARMED BANDITS (MAB)

The work of Krishnamurthy et al. (2024) has used an LLM to sequentially select the arms/actions in MAB and proposed methods to manually design the meta-prompt. Their prompt design consists of 5 components with each having 2 possible choices, which gives rise to a total of $2^5 = 32$ possible prompts. Here we show that our algorithms can be used to automatically optimize their manually designed prompts to further enhance their performance. Specifically, we adopt 2 of their prompt designs: BSSND and BSSCD, and apply our EXPO and EXPO-ES algorithms to optimize the important components in these prompt designs. Following Krishnamurthy et al. (2024), we use two MAB instances: *easy* and *hard*. We adopt GPT-4-Turbo as the LLM here. More details on the experimental design are deferred to App. D.3.1. The results for the 4 experimental settings (i.e., 2 prompt designs × 2 MAB instances) are shown in Fig. 4, which demonstrate that our EXPO and EXPO-ES algorithms are able to significantly reduce the cumulative regret of MAB in this task across different prompt desings and MAB instances. We illustrate the comparison between the original meta-prompt and the one optimized by our EXPO in Figs. 18 and 19 in App. E.5.

# 5 ABLATION STUDY

**Only Optimizing Task Description or Meta-Instruction.** Our `EXPO` jointly optimize the task description $\mathcal{D}$ and the meta-instruction $\mathcal{I}$. Here we evaluate the performance of optimizing either $\mathcal{D}$ or $\mathcal{I}$ alone. The results in Fig. 6 show that jointly optimizing them indeed leads to significantly better performance. However, optimizing these components alone still consistently outperforms OPRO.

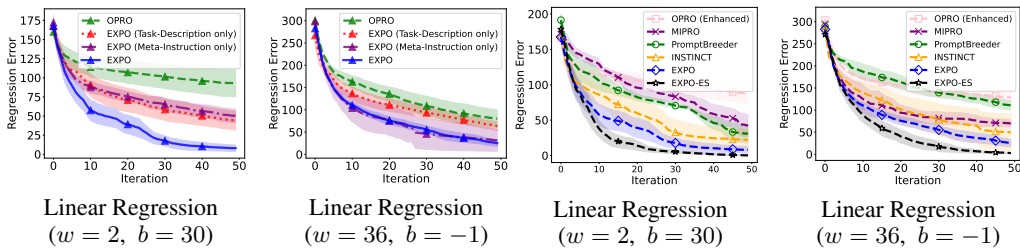

| Linear Regression $(w = 2, \; b = 30)$ | Linear Regression $(w = 36, \; b = -1)$ | Linear Regression $(w = 2, \; b = 30)$ | Linear Regression $(w = 36, \; b = -1)$ |

Figure 6: Results of `EXPO` when *only* optimiz- Figure 7: Comparison of our `EXPO` with general ing the task description or the meta-instruction. prompt optimization methods in the LR tasks.

**Comparison with General Prompt Optimization Methods.** As discussed in Sec. 1, in meta-prompt optimization for LLM-based agents, ***the non-stationary reward observations render the previous general prompt optimization methods unsuitable***. Here we verify this by comparing our `EXPO` with a variety of general prompt optimization methods, including the INSTINCT algorithm based on stochastic MAB (Lin et al., 2024b), PromptBreeder (Fernando et al., 2023) and MIPRO (Opsahl-Ong et al., 2024). The results for the Linear Regression tasks are displayed in Fig. 7, which demonstrate that general prompt optimization methods indeed significantly underperform in the problem of meta-prompt optimization for LLM-based agents. We have also compared with INSTINCT (Lin et al., 2024b) (i.e., the best-performing baseline method from Fig. 7) in the TSP tasks, and the results (Fig. 20 in App. E.6) also show that INSTINCT consistently performs worse than our `EXPO`. These results provide further justifications for our proposed adversarial bandit-based algorithms.

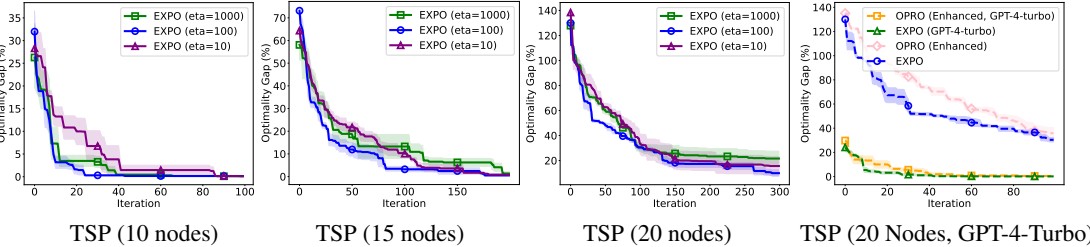

| TSP (10 nodes) | TSP (15 nodes) | TSP (20 nodes) | TSP (20 Nodes, GPT-4-Turbo) |

Figure 8: First three figures: ablation study on impact of exploration parameter $\eta$. Rightmost figure: results using GPT-4-Turbo.

**Impact of the Degree of Exploration.** Here we examine the impact of the degree of exploration, i.e., the value of $\eta$ (line 10 of Algo. 1). The results (Fig. 8, first three figures) show that an excessively large degree of exploration (i.e., a small $\eta = 10$) or an overly small degree of exploration (i.e., a large $\eta = 1000$) both deteriorate the performance. Moreover, the results also demonstrate that in easier tasks (i.e., TSP with 10 nodes), imposing a smaller degree of exploration (i.e., $\eta = 1000$) leads to better performance compared to $\eta = 10$, because it allows our `EXPO` to quickly converge to the optimal solution. Meanwhile, in more challenging tasks (i.e., TSP with 20 nodes), more exploration (i.e., $\eta = 10$) results in better performance (than $\eta = 1000$), because it makes it easier for our `EXPO` to escape local optimum.

**Experiments With Other LLMs.** To evaluate the effectiveness of our approach when combined with different LLMs, here we adopt the challenging TSP task with 20 nodes and replace the GPT-3.5-Turbo model used in our original experiments (Sec. 4.1) by the more advanced GPT-4-Turbo model. The results in Fig. 8 (rightmost figure) show that GPT-4-Turbo significantly improves the performance of both OPRO and our `EXPO`. More importantly, as visualized more clearly in Fig. 14 in App. E.1, when both adopting GPT-4-Turbo, our `EXPO` still significantly outperforms OPRO. The results show that our `EXPO` can effectively improve the performance of LLM-based agents across different LLMs.

**Effectiveness of the Optimal Prompt Discovered by `EXPO`.** To further verify the ability of our `EXPO` to identify effective meta-prompts, here we replace the original task description and meta-instruction in an LLM-based sequential decision-making algorithm (e.g., OPRO) by the optimal ones discovered by our `EXPO`. For example, for ORPO, we firstly run our `EXPO` to completion, and then use the final meta-prompt selected by our `EXPO` as the meta-prompt to execute OPRO again. The results in Fig. 9 show that fixing the meta-prompt to be the one optimized by our `EXPO` leads to dramatic performance boost to LLM-based sequential decision making.

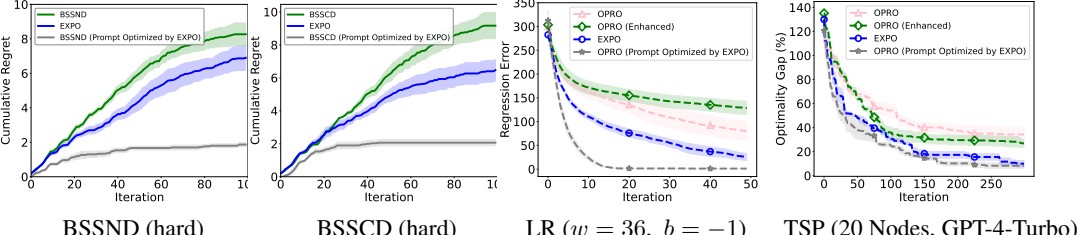

| BSSND (hard) | BSSCD (hard) | LR ($w = 36$, $b = -1$) | TSP (20 Nodes, GPT-4-Turbo) |

Figure 9: Results achieved by fixing the meta-prompt to be the optimal one discovered by our `EXPO` (gray curves).

**Computational Efficiency.** We have also explored the computational efficiency of our `EXPO` algorithm. As shown in Table 1 in Appendix E.4, our `EXPO` requires substantially less computational time than OPRO to achieve the same level of performance.

## 6 RELATED WORK

**Prompt Optimization.** Prompt optimization has attracted increasing attention in recent years. Early studies primarily focused on optimizing prompts for *white-box* LLMs using gradient-based techniques (Shi et al., 2023; Shin et al., 2020), reinforcement learning approaches (Deng et al., 2022), and soft-prompt tuning methods (Lester et al., 2021; Li & Liang, 2021; Zhong et al., 2021). More recently, a growing body of work has investigated prompt optimization for *black-box* LLMs, leveraging evolutionary algorithms (Fernando et al., 2023; Guo et al., 2024; Zhou et al., 2023), zeroth-order optimization (Hu et al., 2024; Zhan et al., 2024), Monte Carlo tree search (Wang et al., 2023b), and the Hyperband algorithm (Schneider et al., 2024). Of note, several methods have introduced multi-armed bandit frameworks (Chen et al., 2023; Lin et al., 2024b; Shi et al., 2024) and reinforcement learning strategies (Kong et al., 2024) to achieve sample-efficient prompt optimization in the black-box setting. Despite these advances, to the best of our knowledge, our algorithm is the first to efficiently optimize the *meta-prompt* for LLM-based agents in sequential decision-making tasks. We defer a discussion of related works on automated exemplar selection to App. A.

**LLM-Based Sequential Decision Making.** Recent studies have explored leveraging LLMs to address sequential decision-making problems, including Bayesian optimization (Yang et al., 2024), multi-armed bandits (Chen et al., 2024; Krishnamurthy et al., 2024; Mukherjee et al., 2024; Xia et al., 2024), and reinforcement learning (Dai et al., 2024; Monea et al., 2024; Wang et al., 2024a). However, most of these approaches rely on a fixed, manually designed meta-prompt for guiding the LLM, which limits their ability to fully exploit the potential of LLM-based sequential decision-making. Meanwhile, the broader field of LLM-based agents has seen rapidly growing interest. A variety of benchmarks have been proposed to evaluate their capabilities, such as AgentBench (Liu et al., 2023), SmartPlay (Wu et al., 2023), and AgentGym (Xi et al., 2024), among others. For a more comprehensive overview of recent progress and open challenges in LLM-based agents, we refer readers to recent surveys (Cheng et al., 2024; Wang et al., 2024b; Xi et al., 2023).

## 7 CONCLUSION

In this work, we propose the `EXPO` algorithm to automatically optimize meta-prompts for LLM-based sequential decision-making tasks. We further extend `EXPO` to develop the `EXPO-ES` algorithm, which also optimizes the exemplars within the meta-prompt. Extensive experiments demonstrate that our proposed algorithms considerably and consistently enhance the performance of LLM-based sequential decision-making. A potential limitation of this work is its inapplicability to optimizing meta-prompts for LLM-based multi-agent systems, an area we plan to explore in future research.

## REPRODUCIBILITY STATEMENT

To ensure the reproducibility of our research, we provide a comprehensive account of our algorithms, experimental configurations, and implementation specifics. The complete, anonymized source code is included as supplementary material. The core logic of our proposed methods, `EXPO` and `EXPO-ES`, is formally delineated with pseudocode in Algo. 1 and Algo. 2, respectively. All requisite details for replicating our experimental results are thoroughly documented. Appendix D.1 specifies the methodology for generating the prompt domain. Exhaustive settings for the Linear Regression, TSP, and Instruction Optimization tasks are detailed in Appendix D.2, which covers evaluation metrics, prompt scoring functions, model choices, and key hyperparameters. Similarly, all configurations for the Multi-Armed Bandit experiments are specified in Appendix D.3. To guarantee statistical robustness, all experiments were repeated multiple times, with the exact number of repetitions explicitly stated in the relevant appendix sections. Furthermore, critical implementation details, such as numerical stability, and computational resources (Appendix D.5) are provided.

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

## A    ADDITIONAL RELATED WORK

**Automated Exemplar Selection.**    Another line of research explores automatic exemplar selection for in-context learning. Some methods (Li & Qiu, 2023; Nguyen & Wong, 2023; Wang et al., 2023a; Zhang et al., 2022) select a single static set of exemplars for all test queries within a task, whereas others (Albalak et al., 2024; Gao et al., 2024; Gupta et al., 2023; Levy et al., 2023; Liu et al., 2022; Rubin et al., 2022; Ye et al., 2023) train a retriever model to dynamically select the most relevant set of exemplars for each query. Furthermore, a few recent approaches (Opsahl-Ong et al., 2024; Wan et al., 2024; Wu et al., 2024) have achieved joint prompt optimization and exemplar selection.

## B    THE USE OF LARGE LANGUAGE MODELS (LLMS)

We used a Large Language Model (LLM) to assist with polishing the writing and refining the phrasing in this paper.

## C    OUR EXPO-ES ALGORITHM TO ADDITIONALLY OPTIMIZE THE EXEMPLAR SEQUENCES

---

**Algorithm 2** EXPO-ES

---

**input** Initial task description $\mathcal{D}_0$, initial meta-instruction $\mathcal{I}_0$. Maximum number $\mathcal{L}$ of exemplars in the meta-prompt, the number $k^{\text{ES}}$ of exemplar sequences in the domain.

1:  Initialize the exemplar set $\mathcal{E}_0 = \emptyset$, and the subset $\mathcal{E}_0' = \emptyset$, meta-prompt-score set $\mathcal{S}_0 = \emptyset$, and cumulative score estimates $\hat{s}_j^{(0)}$ for all $j \in \{1, \ldots, k^{\text{ES}}\}$. Initialize the history of NN parameters $\Theta_{\text{history}} = \emptyset$, and the exemplar training set $\mathcal{T}_0 = \emptyset$.

2:  **for** iteration $t = 0, 1, \ldots, T - 1$ **do**

3:     **Lines 3-11 of Algo. 1**.

4:     Compute the embedding $g(\mathcal{E}_t')$ of the selected exemplar sequence $\mathcal{E}_t'$, and add $g(\mathcal{E}_t')$ and its score $s_t$ to the exemplar training set: $\mathcal{T}_{t+1} \leftarrow \mathcal{T}_t \cup \{(g(\mathcal{E}_t'), s_t)\}$.

5:     Update the parameters $\theta^{\text{ES}}$ of the NN $\mathcal{M}_{\text{ES}}(g(\cdot); \theta^{\text{ES}})$ by using the updated $\mathcal{T}_{t+1}$ as the training set to minimize the MSE loss, yielding $\theta_{t+1}^{\text{ES}}$.

6:     Add the updated parameters to the history: $\Theta_{\text{history}} \leftarrow \Theta_{\text{history}} \cup \{\theta_{t+1}^{\text{ES}}\}$.

7:     **if** $|\mathcal{E}_{t+1}| > \mathcal{L}$ **then**

8:        Randomly generate $k^{\text{ES}}$ sequences of $\mathcal{L}$ exemplars from the exemplar set $\mathcal{E}_{t+1}$: $\{\mathcal{E}_{t+1}^1, \mathcal{E}_{t+1}^2, \ldots, \mathcal{E}_{t+1}^{k^{\text{ES}}}\}$, in which every $\mathcal{E}_{t+1}^j$ represents an ordered set of $\mathcal{L}$ exemplars from $\mathcal{E}_{t+1}$.

9:        Initialize cumulative score estimates $\hat{s}_j^{(0)} = 0$ for all $j \in \{1, \ldots, k^{\text{ES}}\}$.

10:        **for** each $\mathcal{E}_{t+1}^j$ in $\{\mathcal{E}_{t+1}^1, \ldots, \mathcal{E}_{t+1}^{k^{\text{ES}}}\}$ **do**

11:           Initialize cumulative score $\hat{s}_j^{(0)} = 0$.

12:           **for** each historical model parameter $\theta_i^{\text{ES}} \in \Theta_{\text{history}}$ **do**

13:              Update the cumulative score for $\mathcal{E}_{t+1}^j$: $\hat{s}_j^{(i)} = \hat{s}_j^{(i-1)} + \mathcal{M}_{\text{ES}}(g(\mathcal{E}_{t+1}^j); \theta_i^{\text{ES}})$.

14:        Compute the final cumulative score estimates: $\hat{s}_j^{(\text{final})} = \hat{s}_j^{(|\Theta_{\text{history}}|)}, \quad \forall j \in \{1, \ldots, k^{\text{ES}}\}$.

> cumulative score estimates

15:        Compute the sampling distribution $P_t^{\text{ES}}$ over the $k$ exemplar sequences:

$$P_t^{\text{ES}}[j] = \frac{\exp(\eta \hat{s}_j^{(\text{final})})}{\sum_{l=1}^{k^{\text{ES}}} \exp(\eta \hat{s}_l^{(\text{final})})}, \quad \forall j \in \{1, \ldots, k^{\text{ES}}\}. \tag{5}$$

16:        Sample an exemplar sequence $\mathcal{E}_{t+1}' \sim P_t^{\text{ES}}$.

---

Our complete EXPO-ES algorithm is described in Algo. 2. As we have discussed in Sec. 3.2, there are two major differences compared to the way in which our EXPO algorithm optimizes the task description and meta-instruction (Algo. 1). Firstly, our domain of $k^{\text{ES}}$ arms (i.e., every arm

corresponds to a randomly sampled exemplar sequence) changes in every iteration (line 8). Secondly, as a result of the time-varying domains, we need to save a copy of the parameters of the NN trained in every iteration in order to compute the cumulative score estimates (lines 9-14).

**Simplified Variant of Our `EXPO-ES` Algorithm.** When applying our `EXPO-ES` algorithm to the LLM-based MAB algorithm in Krishnamurthy et al. (2024) (Sec. 4.2), we have adopted a simplified variant of our `EXPO-ES`. This is because in the problem setting from Krishnamurthy et al. (2024), the number of arms is small. Therefore, instead of including a subset of the history of exemplars in the prompt, their algorithm has instead included a summarized observation history. An example of such summarized observation history with 5 arms (represented by 5 buttons with different colors) is given in Fig. 10 below. Therefore, here we aim to optimize the format of the summarized observation history. Specifically, we *optimize the order of the arms* in the summarized history, and our domain of arms consist of *all cyclically shifted variants of the following sequence of buttons*: {blue button, green button, red button, yellow button, purple button}. For example, some other arms (button sequences) in our domain include: {green button, red button, yellow button, purple button, blue button} and {red button, yellow button, purple button, blue button, green button}. As a result, unlike our original `EXPO-ES` algorithm described in Sec. 3.2, here we do not suffer from the issue of *time-varying domain of arms*.

---

...

blue button: pressed 2 times with average reward 0.5
green button: pressed 1 times with average reward 0.0
red button: pressed 1 times with average reward 1.0
yellow button: pressed 0 times
purple button: pressed 1 times with average reward 0.0

...

---

Figure 10: An example of the summarized observation history used by the LLM-based MAB algorithm from Krishnamurthy et al. (2024).

Therefore, when applying our `EXPO-ES` algorithm to improve the LLM-based MAB method from Krishnamurthy et al. (2024) (Sec. 4.2), we make two modifications to our standard `EXPO-ES` algorithm described in Algo. 2. Firstly, instead of randomly sampling $k^{\text{ES}}$ exemplar sequences to form our domain of exemplar sequences, here our domain remains fixed across different iterations, i.e., all cyclically shifted variants of the arms. Secondly, since here we do not suffer from the issue of time-varying domain of arms (i.e., exemplar sequences), we can resort to the incremental update of the cumulative reward estimates adopted by our `EXPO` algorithm (line 9 of Algo. 1). As a result, we do not need to save a copy of the parameters of the NN trained in every iteration.

## D  MORE DETAILS ON OUR EXPERIMENTAL SETTINGS

### D.1  MORE DETAILS ON THE GENERATION OF THE DOMAIN OF TASK DESCRIPTION AND META-INSTRUCTION

Here we describe the details about how we generate the domain of task descriptions and meta-instructions. Below we provide the prompt we have used to instruct the LLM to generate every prompt in the domain.

---

**Example Query: Meta-Prompt Instruction Rephrasing Template**

To achieve a more effective TASK description and INSTRUCTION and convey its core essence more clearly, please enhance the content in the quote by rephrasing and changing some information:
"{INITIAL_META-PROMPT}"
Please return directly the modified description without additional description.
The modified description:

---

**Generation of the Domain.** To effectively generate task-specific prompts, we utilized an initial prompt to guide the LLM in creating diverse task descriptions and meta-instructions. For each task, the LLM was prompted 100 times to rephrase the task description and meta-instruction separately, resulting in 100 unique rephrased prompts for each. Combined with the initial prompt, this process produced a total of $101 \times 101$ combinations of task descriptions and meta-instructions for each task.

To optimize computational efficiency, we pre-compute the embeddings of all task descriptions and meta-instructions in the domain using the embedding model $g(\cdot)$ and store the results to prevent redundant calculations during subsequent experiments.

For the rephrasing process, we employed the GPT-4 model with a temperature setting of 1.3, ensuring diverse and high-quality rephrased prompts for both task descriptions and meta-instructions.

### D.2 MORE DETAILS ON OPRO FOR THE LINEAR REGRESSION, TRAVELING SALESMAN PROBLEM AND INSTRUCTION OPTIMIZATION (SEC. 4.1)

#### D.2.1 TASK SETTING.

**Linear Regression.** We conduct experiments on Linear Regression by selecting two challenging ground truth weight-bias $(w, b)$ pairs. The experiments follow the OPRO framework, which requires warm-starting the LLM with initial exemplars. Using a fixed random seed, we first generate 50 random data points uniformly distributed within the range $[-1, 1]$, which perfectly satisfy the ground truth $w_{\text{true}}, b_{\text{true}}$ pairs, ensuring that these data points can serve as the foundation for evaluating the LLM's ability to model the relationships. Additionally, 5 $w, b$ pairs with corresponding scores, sampled within the range $[10, 20]$, are generated using another fixed random seed to serve as the initial exemplars. At each iteration, the LLM is prompted 8 times (consisting of 1 inference with a temperature setting of $T = 0$ and 7 inferences with a temperature setting of $T = 1$) using the current exemplars, and the prompt is updated based on the generated outputs. The exemplars are dynamically updated to include the top 20 $w, b$ pairs and their associated scores from all historical records across iterations, ensuring the LLM is always guided by the best-performing examples. The total number of iterations is set to 50, and each ground truth configuration is repeated 5 times for consistency.

**Traveling Salesman Problem (TSP).** For the TSP task, experiments are conducted on three problem sizes defined by the number of nodes: 10, 15, and 20. For each TSP instance, the problem is defined by randomly generating $n = 10, 15, 20$ nodes, where the $x$ and $y$ coordinates of each node are sampled uniformly from the range $[-100, 100]$. For each configuration, a specific TSP instance is generated using a fixed random seed, and a single random seed is used to generate warm-start exemplars to initialize the LLM prompts. To initialize the optimization process, we randomly sample 5 different TSP routes along with their corresponding total distances. These routes and their lengths are used as the initial exemplars for the LLM. Each iteration consists of 8 prompt calls to the LLM, followed by an update of the exemplars based on the generated results. More specifically, during each iteration, the GPT-3.5-turbo is prompted 8 times using the same prompt, consisting of 1 inference with a temperature setting of $T = 0$ to ensure stability and 7 inferences with a temperature setting of $T = 1$ to encourage exploration. Similar to the Linear Regression task, the exemplars for TSP are updated to include the top 20 historical solutions with the best scores, ensuring the prompt leverages the most effective examples. The number of iterations is set to 100, 200, and 300 for 10-node, 15-node, and 20-node TSP problems, respectively, to account for the increasing complexity of the tasks. Each node configuration is repeated 3 times to ensure consistency and reliability.

**Instruction Optimization.** Following the experimental settings in OPRO, we have conducted instruction optimization experiments using the GSM8K benchmark dataset (Cobbe et al., 2021). In line with OPRO, which emphasizes sample-efficient optimization, we randomly select 75 problem–answer

pairs from the GSM8K training set as the instruction tuning subset. GPT-4-Turbo is used as the instruction optimizer to iteratively generate improved instructions, while Qwen-2.5-7B serves as the scorer LLM to evaluate the effectiveness of the instructions. During each iteration, GPT-4-Turbo is prompted 8 times with the current set of exemplars: 1 inference is performed with temperature $T = 0$ and the other 7 inferences are conducted with $T = 1$, yielding 8 candidate instructions per round. Each candidate is evaluated using the `Q_end` strategy by prompting Qwen-2.5-7B on the 75 training questions. The exemplar set for the next iteration consists of the 20 best instructions based on evaluation scores, along with 3 randomly sampled training exemplars. Here our `EXPO` algorithm only considers optimizing the meta-instruction component in the meta-prompt, while keeping the task description fixed. This is because in the setting of the instruction optimization experiments in OPRO, the task description is instantiated as a set of three randomly selected exemplars from the training set, which makes it not directly amenable to the application of our meta-prompt optimization algorithm. As such, only the meta-instruction is optimized in this experiment. The optimization is initialized with the Chain-of-Thought (CoT) prompt prefix "Let's think step by step", which serves as the initial instruction for OPRO, `EXPO` and `EXPO-ES`. Each experiment is repeated 3 times.

### D.2.2 EVALUATION METRICS.

**Linear Regression.** In the Linear Regression task, the performance of the algorithms is evaluated using the Mean Squared Error (MSE) metric. Given a set of $N$ one-dimensional input data points $\mathbf{x} \in \mathbb{R}$ and their corresponding ground truth labels $\mathbf{y} \in \mathbb{R}$, the MSE is computed as:

$$\text{MSE} = \frac{1}{N} \left\| \mathbf{y} - (w \cdot \mathbf{x} + b) \right\|^2,$$

where $w \in \mathbb{R}$ and $b \in \mathbb{R}$ are the weight and bias parameters inferred by the LLM, and $N$ is the total number of data points.

**Traveling Salesman Problem (TSP).** For the TSP task, the performance of the LLM-generated solutions is evaluated based on the total Euclidean distance of the TSP tour. Given a set of two-dimensional points $\{(x_i, y_i)\}_{i=1}^{N}$, where $N$ is the total number of nodes, the length of a proposed TSP tour $\mathcal{P} = [\pi(1), \pi(2), \ldots, \pi(N), \pi(1)]$ is computed as:

$$\text{Length} = \sum_{i=1}^{N} \sqrt{\left( x_{\pi(i+1)} - x_{\pi(i)} \right)^2 + \left( y_{\pi(i+1)} - y_{\pi(i)} \right)^2},$$

where $\pi$ represents the permutation of nodes in the proposed tour, and $\pi(N+1) = \pi(1)$ ensures the tour returns to the starting node.

To evaluate the convergence and effectiveness of the agents, we use the *Optimality Gap* metric, which quantifies the deviation of the solver's best-found solution from the true optimal solution. It is defined as:

$$\text{Optimality Gap} = \frac{\text{SolverOptimal} - \text{Optima}}{\text{Optima}} \times 100\%,$$

where:

- SolverOptimal denotes the shortest tour length found by the solver up to the current iteration.
- Optima is the length of the known optimal TSP tour.

**Instruction Optimization.** To assess the effectiveness of each generated instruction, we adopt the `Q_end` evaluation strategy introduced in OPRO. For a given instruction, prompts are constructed by appending the instruction to each question in the set, and the Qwen-2.5-7B-Instruct model with the temperature set to 0 is queried to generate corresponding answers. The final predicted outputs are then compared to ground-truth answers using exact numerical match accuracy. Specifically, for the GSM8K dataset, an answer is considered correct if and only if it exactly matches the reference solution in its numerical form—partial matches, approximations, or differently formatted correct values are all treated as incorrect.

Formally, the score is defined as the average accuracy over the set:

$$\text{Accuracy} = \frac{1}{N} \sum_{i=1}^{N} \mathbb{1}[\hat{y}_i = y_i], \tag{2}$$

where $\hat{y}_i$ denotes the model-predicted answer for the $i$-th example, $y_i$ is the corresponding ground-truth answer, and $\mathbb{1}[\cdot]$ is the indicator function that equals 1 when the predicted and true answers exactly match. Higher average accuracy reflects that the algorithm has identified more effective instructions, indicating greater optimization efficiency and overall method quality.

### D.2.3 DESIGN OF PROMPT SCORE.

**Linear Regression and TSP.** In both the Linear Regression and TSP tasks, optimal solutions are characterized by lower evaluation scores. To align with the requirements of the algorithm and ensure more stable learning, we define the *Meta-Prompt Score* using the formula:

$$\text{Meta-Prompt Score} = \frac{-\text{Evaluation Score} + b}{b},$$

where $b > 0$ is a stabilizing constant. This formulation ensures that lower evaluation scores correspond to higher prompt scores, which better facilitates the optimization process and contributes to steady algorithmic learning.

For the Linear Regression task, the *Evaluation Score* is defined as the Mean Squared Error (MSE) of the weight-bias $(w, b)$ pairs proposed by the algorithm at each iteration under a Temperature=0 stable inference. The MSE is computed based on the provided one-dimensional data points.

For the TSP task, the *Evaluation Score* corresponds to the total Euclidean distance (*Length*) of the TSP tour proposed by the algorithm at each iteration, also under a Temperature=0 stable inference.

**Instruction Optimization.** Instruction Optimization task uses accuracy as the evaluation metric, where higher scores signify superior instruction quality and meta-prompt quality. Accordingly, we directly define the *Meta-prompt Score* as the average accuracy (scaled to $[0, 1]$) achieved by the instruction generated under the current meta-prompt.

Specifically, at each iteration, we perform a Temperature=0 inference with GPT-4-Turbo to generate a candidate instruction from the current meta-prompt. This instruction is then evaluated on the training subset using the Qwen-2.5-7B-Instruct model under the `Q_end` setting, and the resulting average accuracy serves as the Prompt Score:

$$\text{Meta-Prompt Score} = \text{Accuracy}, \tag{3}$$

This formulation directly reflects the effectiveness of the meta-prompt: higher Accuracy values lead to higher Meta-Prompt Scores, indicating more successful instruction optimization.

### D.2.4 DETAILS ABOUT THE MODELS AND PARAMETERS IN OUR ALGORITHMS

**LLM Agents and Embedding Model.** In our experiments, the primary LLM agent used is GPT-3.5-Turbo. For embedding generation, we utilized OpenAI's `text-embedding-3-large` model, which outputs embeddings of dimensionality 3072. These embeddings were used to represent both the task description and meta-instruction in the `EXPO` framework. The embeddings were also employed to represent the exemplars in the `EXPO-ES` framework. During each iteration of inference, the LLM agent performed 1 prediction with a temperature setting of $T = 0$ to provide a stable solution and 7 additional predictions with a temperature setting of $T = 1$ to encourage exploration.

**Neural Network Parameters.** For the Linear Regression and TSP experiments, the input to the neural network consists of the concatenated embeddings of the task description and meta-instruction, resulting in an input dimensionality of $3072 + 3072 = 6144$. The neural network employs a single hidden layer with a width of 1536 and produces a single scalar output. The training objective is to minimize the Mean Squared Error (MSE) loss function.

For the Instruction Optimization experiments, we only optimize the meta-instruction without optimizing the task description. Consequently, the neural network input dimensionality for IO is 3072, corresponding solely to the meta-instruction embedding. The neural network in the IO experiment employs a single hidden layer with a width of 512 and outputs a single scalar value. The training objective is also to minimize the MSE loss.

For the `EXPO-ES`, the exemplar selection process differs depending on the iteration count. During the initial iterations, when fewer than 20 optimal historical records are available, we use

all available exemplars. As the iteration count increases, exemplars are selected from the top $\min(\text{total exemplar records}, 30)$ historical optimal records. From this pool, 257 exemplars are constructed, consisting of 256 randomly selected exemplars and 1 heuristic exemplar generated from a combination of 20 best historical records. The neural network for `EXPO-ES` operates on an input dimensionality of 3072, corresponding to the embedding of a single exemplar. It employs a single hidden layer with a width of 512 and produces a single scalar output. The training objective is to minimize the Mean Squared Error (MSE) loss.

**EXP3 Learning Rate.** In the `EXPO`, the learning rate parameter $\eta_{\text{desc}}$ is set to 100 for selecting task description and meta-instruction combinations in both the Linear Regression and TSP experiments. For the Instruction Optimization experiments, where only the meta-instruction is optimized, the learning rate parameter $\eta_{\text{meta-inst}}$ is set to 50.

In the `EXPO-ES`, the learning rate parameter for selecting combinations of task description and meta-instruction is set identically to that in `EXPO`. And the learning rate parameter for exemplar selection, $\eta_{\text{exemplar}}$, is set to 25.

### D.2.5 ENHANCED OPRO

Here, we describe how we have enhanced the original algorithm (Yang et al., 2024) by modifying its prompts.

During initial experiments with the meta-prompts provided by the original OPRO algorithm (Yang et al., 2024) for task description rephrasing, we observed that the LLM often misinterprets the *descending order* semantics described in the original design. In tasks like TSP and Linear Regression, where better solutions correspond to lower evaluation scores, *descending order* is intended to arrange solutions from high evaluation scores to low. However, the LLM frequently misunderstands this as a *descending order of solution quality*, interpreting higher-ranked solutions as better and lower-ranked ones as worse, which is contrary to the intended meaning.

To address this issue, we enhance the orginal meta-prompts by explicitly clarifying the semantics of descending order in the context of evaluation scores. This modification ensures that the LLM accurately understand the intended instructions. When tested with the enhanced prompts, the problem was resolved, and the LLM is able to consistently generate correct rephrased task descriptions. For a clearer illustration, we provide below the original OPRO meta-prompt (Fig. 11) and our enhanced OPRO meta-prompt (Fig. 12).

---

**The task description in the original OPRO prompt**

You are given a list of points with coordinates below: {POINTS}.
Below are some previous traces and their lengths. The traces are arranged in descending order based on their lengths, where lower values are better.
......

---

Figure 11: The task description in the original OPRO prompt.

---

**The task description in our enhanced OPRO prompt**

You are given a list of points with coordinates below: {POINTS}.
Below are some previous traces and their lengths. The traces are arranged in descending order based on their lengths, where smaller lengths indicate better solutions. Therefore, the traces are listed from the largest length to the smallest, the trace with the smallest length is considered the most optimal.
......

---

Figure 12: The task description in the enhanced OPRO prompt. The texts we have modified are highlighted in red.

### D.3 More Details on the LLM-Based Multi-Armed Bandits Task (Sec. 4.2)

#### D.3.1 Explanation of BSSCD and BSSND

We provide a detailed explanation and demonstration of prompt designs for both BSSCD and BSSND (Krishnamurthy et al., 2024), highlighting their key components and structures. Figure 13 illustrates a complete example of a BSSCD prompt designed for the MAB problem under the hard difficulty setting. It showcases the structure and color-coded components of the prompt in detail.

---

**The setting of the prompt in MAB**

[SYSTEM]
You are a bandit algorithm in a room with 5 buttons labeled blue, green, red, yellow, purple. Each button is associated with a Bernoulli distribution with a fixed but unknown mean; the means for the buttons could be different. For each button, when you press it, you will get a reward that is sampled from the button's associated distribution. You have 100 time steps and, on each time step, you can choose any button and receive the reward. Your goal is to maximize the total reward over the 100 time steps.

At each time step, I will show you a summary of your past choices and rewards. Then you must make the next choice. You may output a distribution over the 5 buttons formatted EXACTLY like "blue:a,green:b,red:c,yellow:d,purple:e". Let's think step by step to make sure we make a good choice.

You must provide your final answer within the tags <Answer>DIST<\Answer> where DIST is the distribution in the format specified above.

---

[USER]
So far you have played 5 times with your past choices and rewards summarized as follows:
blue button: pressed 2 times with average reward 0.5
green button: pressed 1 times with average reward 0.0
red button: pressed 1 times with average reward 1.0
yellow button: pressed 0 times
purple button: pressed 1 times with average reward 0.0
Which button will you choose next? Remember, YOU MUST provide your final answer within the tags <Answer>DIST<\Answer> where DIST is formatted like "blue:a,green:b,red:c,yellow:d,purple:e".

---

Figure 13: A complete example of the prompt in MAB. The different components in the prompt are explained in detail in App. D.3.1.

- **B**utton scenario and **S**uggestive framing, providing the foundational task scenario, clarifying the role of the agent, and framing the objective of the task in a suggestive manner to guide decision-making.
- Description of the multi-armed bandit problem, offering the agent a detailed task description, including comprehensive information about the task's objectives, constraints, and operational details.
- **S**ummarized history, presenting a condensed version of historical decisions and reward feedback to the agent, instead of providing step-by-step decision and reward feedback.
- **C**hain-of-thought or **N**o CoT, indicating whether to encourage the agent to engage in step-by-step reasoning for decision making.
- **D**istribution over actions, encouraging the agent to generate a probability distribution over the arms of the bandit, instead of making deterministic decisions.

When we use our EXPO algorithm to optimize the task description and meta-instruction, the upper section with light purple background corresponds to the Task Description, where as the section below it with light blue background represents the Meta-Instruction. In other words, our EXPO algorithm is used to optimize the text in these two sections.

### D.3.2 TASK SETTING

The experiments are conducted for both the BSSND and BSSCD prompts under two pre-defined difficulty levels: *hard* and *easy*. For the *hard* setting, the MAB instance consists of $K = 5$ arms, where the best arm has a mean reward of $\mu^\star = 0.5 + \Delta/2$ with $\Delta = 0.2$, and all other arms have a mean reward of $\mu = 0.5 - \Delta/2$. For the *easy* setting, the MAB instance consists of $K = 4$ arms with a larger gap $\Delta = 0.5$ between the best arm and the suboptimal arm. We set the blue button as the optimal arm in experiments, corresponding to the arm with the highest expected reward. Each configuration is tested using two fixed random seeds, with experiments repeated 3 times for each seed, resulting in a total of $2 \times 3 = 6$ runs per setting. Each experiment consists of 100 iterations, with the LLM-based agents making decisions and updating prompts iteratively to optimize performance. The work of Krishnamurthy et al. (2024) has reported that GPT-3.5 models encounter exploration failures in MAB tasks, making them unsuitable as agents for solving such problems. In contrast, GPT-4 demonstrates the capability to effectively handle the exploration-exploitation trade-off inherent in MAB settings. Therefore, we adopt GPT-4-turbo as the LLM agent for this experiment.

### D.3.3 EVALUATION METRIC.

In the LLM-based Multi-Armed Bandit (MAB) task (Sec. 4.2), the performance of the LLM agent is assessed using the *Cumulative Regret* metric. At each iteration, the LLM agent outputs a probability distribution over the arms, representing the likelihood of sampling each arm.

Formally, let there be $K$ arms, each associated with an expected reward $\mu_1, \mu_2, \ldots, \mu_K$, where $\mu^* = \max_{k \in \{1,\ldots,K\}} \mu_k$ denotes the expected reward of the optimal arm. At iteration $t$, we sample an arm $a_t \in \{1, \ldots, K\}$, which is determined by the probability distribution provided by the LLM agent. The instantaneous regret for iteration $t$ is then defined as:

$$r_t = \mu^* - \mu_{a_t},$$

where $\mu_{a_t}$ represents the expected reward of the selected arm $a_t$ at iteration $t$.

The cumulative regret after $T$ iterations is computed as:

$$R_T = \sum_{t=1}^{T} r_t = \sum_{t=1}^{T} \left( \mu^* - \mu_{a_t} \right).$$

### D.3.4 DESIGN OF PROMPT SCORE.

The score of the prompt is designed to quantify the expected reward of the LLM agent's sampling strategy at each iteration. At iteration $t$, the LLM agent outputs a sampling probability distribution $\{p_1, p_2, \ldots, p_K\}$, where $p_i$ represents the probability of selecting arm $i$ ($i = 1, 2, \ldots, K$, with $K$ being the total number of arms). Simultaneously, the historical records from the first $(t-1)$ iterations allow us to compute an unbiased estimate of the Bernoulli reward parameter for each arm, $\hat{\mu}_i$, based on the observed rewards and sampling counts.

For arm $i$, the Bernoulli parameter $\hat{\mu}_i$ is estimated as:

$$\hat{\mu}_i = \begin{cases} 0, & \text{if } n_i = 0, \\ \frac{\sum_{j=1}^{t-1} R_{i,j}}{n_i}, & \text{if } n_i > 0. \end{cases}$$

where $\sum_{j=1}^{t-1} R_{i,j}$ denotes the cumulative reward obtained from arm $i$ during the first $(t-1)$ iterations, and $n_i$ represents the total number of times arm $i$ was sampled during the same period.

The LLM agent's expected reward $\hat{R}_{\text{expected}}$ at iteration $t$ is then calculated by weighting the estimated Bernoulli parameters $\{\hat{\mu}_1, \hat{\mu}_2, \ldots, \hat{\mu}_K\}$ with the sampling probabilities $\{p_1, p_2, \ldots, p_K\}$ provided by the LLM:

$$\hat{R}_{\text{expected}} = \sum_{i=1}^{K} p_i \cdot \hat{\mu}_i.$$

This expected reward $\hat{R}_{\text{expected}}$ serves as the score of the prompt.

**Motivation for the Score Design.** The design of the prompt score is driven by the objective of guiding the LLM agent to favor arms with higher expected rewards, represented by $\mu_i$. Since the true values of $\mu_i$ are not available, the prompt score is designed to estimate this quantity based on observed data. Specifically, the higher the value of $\mu_i$, the higher the sampling probability $p_i$ should be assigned to arm $i$, reflecting the optimal choice. Conversely, arms with lower values of $\mu_i$ should be assigned lower probabilities.

The original score with the Bernoulli parameters:

$$R_{\text{expected}} = \sum_{i=1}^{K} p_i \mu_i.$$

In the absence of the true $\mu_i$, we rely on the unbiased estimates $\hat{\mu}_i$:

$$\hat{R}_{\text{expected}} = \sum_{i=1}^{K} p_i \hat{\mu}_i.$$

This design is justified because, for most of iterations, the score $\sum_{i=1}^{K} p_i \hat{\mu}_i$ is an unbiased estimate of the true expected reward $\sum_{i=1}^{K} p_i \mu_i$, and we proceed to formally establish this unbiasedness.

**Proof of Unbiasedness.** For iteration $t$, where $n_i > 0$ for all $i$, we aim to show that the score $\sum_{i=1}^{K} p_i \hat{\mu}_i$ is an unbiased estimate of the true expected reward $\sum_{i=1}^{K} p_i \mu_i$. Since $\hat{\mu}_i$ is an unbiased estimate of $\mu_i$, we have:

$$\mathbb{E}\left[\hat{\mu}_i\right] = \mathbb{E}\left[\mu_i\right],$$

Thus, by the linearity of expectation, we obtain:

$$\mathbb{E}\left[\sum_{i=1}^{K} p_i \hat{\mu}_i\right] = \sum_{i=1}^{K} p_i \mathbb{E}\left[\hat{\mu}_i\right]$$

$$= \sum_{i=1}^{K} p_i \mathbb{E}\left[\mu_i\right]$$

$$= \mathbb{E}\left[\sum_{i=1}^{K} p_i \mu_i\right]$$

This shows that the score $\hat{R}_{\text{expected}}$ is an unbiased estimate of the true expected reward $R_{\text{expected}}$.

### D.3.5 DETAILS ABOUT THE MODELS AND PARAMETERS IN OUR ALGORITHMS

**LLM Agents and Embedding Model.** For the MAB tasks, the primary LLM agent is GPT-4-Turbo and the fixed inference temperature is set to $T = 0$. For embedding generation, we employed OpenAI's `text-embedding-3-large` model, which outputs embeddings with a dimensionality of 3072. These embeddings are utilized to represent the prompts provided to the LLM agent during the experiments. At each iteration, the LLM is prompted once using the designed prompt.

**Neural Network Parameters.** For the EXPO, the input to the neural network consists of the concatenated embeddings of the task description and meta-instruction, resulting in an input dimensionality of $3072 + 3072 = 6144$. The neural network employs a single hidden layer with a width of 1536 and produces a scalar output. The model is trained by minimizing the Mean Squared Error (MSE) loss function.

For the EXPO-ES, the neural network is designed to process $K$ exemplars, where $K$ is determined by the total number of available summaries. To ensure fairness, $K$ distinct exemplar combinations are generated at each iteration using a cyclic rotation mechanism. This mechanism ensures that each summary occupies every possible position within the exemplar sequence. Formally, given $K$ summaries indexed as $\{e_0, e_1, \ldots, e_{K-1}\}$, the $i$-th exemplar combination is defined as:

$$\left(e_i, e_{(i+1) \mod K}, \ldots, e_{(i+K-1) \mod K}\right).$$

This guarantees that each summary appears in every position across all $K$ combinations.

Each exemplar is embedded into a 3072-dimensional vector using the embedding model, and these embeddings are processed individually by the neural network. The neural network consists of a single hidden layer with a width of 512 and produces a scalar output. Like EXPO, the training objective is to minimize the Mean Squared Error (MSE) loss function.

**EXP3 Learning Rate.** For the EXPO, the learning rate parameter $\eta_{\text{desc}}$ is set to 10 for selecting task description and meta-instruction combinations. In the EXPO-ES, two learning rate parameters are used: $\eta_{\text{desc}}$ is set to 10 for selecting task description and meta-instruction combinations, and $\eta_{\text{exemplar}}$ is set to 10 for selecting exemplars.

## D.4 IMPROVING NUMERICAL STABILITY

To prevent numerical overflow during the computation of exponentials in our algorithms, a translation constant $C^{(t)}$ is introduced at each iteration $t$. This constant stabilizes the computation by shifting the cumulative scores, ensuring the algorithm operates reliably until convergence without altering the resulting probability distribution. The translation constant is defined as:

$$C^{(t)} = \max_j S_j^{(t)}. \tag{4}$$

The translated scores are:

$$\tilde{S}_i^{(t)} = S_i^{(t)} - C^{(t)}. \tag{5}$$

The probability distribution after translation is:

$$\tilde{P}_t[i] = \frac{\exp\left(\eta \tilde{S}_i^{(t)}\right)}{\sum_{j=1}^k \exp\left(\eta \tilde{S}_j^{(t)}\right)}. \tag{6}$$

Substituting $\tilde{S}_i^{(t)} = S_i^{(t)} - C^{(t)}$:

$$\tilde{P}_t[i] = \frac{\exp\left(\eta\left(S_i^{(t)} - C^{(t)}\right)\right)}{\sum_{j=1}^k \exp\left(\eta\left(S_j^{(t)} - C^{(t)}\right)\right)}. \tag{7}$$

Using $\exp(a - b) = \frac{\exp(a)}{\exp(b)}$:

$$\tilde{P}_t[i] = \frac{\exp\left(\eta S_i^{(t)}\right) / \exp\left(\eta C^{(t)}\right)}{\sum_{j=1}^k \left(\exp\left(\eta S_j^{(t)}\right) / \exp\left(\eta C^{(t)}\right)\right)}. \tag{8}$$

Simplifying:

$$\tilde{P}_t[i] = \frac{\exp\left(\eta S_i^{(t)}\right)}{\sum_{j=1}^k \exp\left(\eta S_j^{(t)}\right)}. \tag{9}$$

Thus, the probabilities remain unchanged:

$$\tilde{P}_t[i] = P_t[i]. \tag{10}$$

## D.5 COMPUTE RESOURCES AND LICENSES

All experiments were conducted on a server equipped with an NVIDIA L4 GPU and 53 GB of RAM. The GSM8K dataset (Cobbe et al., 2021) and the implementations of the baseline methods of PromptBreeder (Fernando et al., 2023) and MIPRO (Opsahl-Ong et al., 2024) are all under the MIT license.

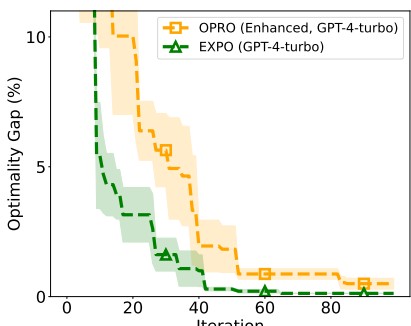

Figure 14: Ablation study results using GPT-4-turbo in the TSP task with 20 nodes.

# E    MORE EXPERIMENTAL RESULTS

## E.1    RESULTS OF GPT-4-TURBO FOR TSP

Fig. 14 shows a zoomed version of Fig. 8 (bottom right) in the main paper. It shows that when GPT-4-Turbo is used as the LLM, our EXPO is still able to significantly outperform OPRO.

## E.2    RESULTS OF OTHER VARIANTS OF OPRO

As we have discussed in Sec. 3.1 and Sec. 4.1, the original OPRO uses a temperature of 1 to choose all 8 actions in a batch, while we have made a slight modification such that we choose the last action in the batch with a temperature of 0. Here we show that this has a minimal impact on the performance of OPRO (Fig. 15). Specifically, in Fig. 15, the orange curves represent the original OPRO (using a temperature of 1 for all 8 actions) and the pink curves correspond to our modified version. We have also compared the performances of the enhanced variants (see Sec. 4.1 for details) for both the original (purple) and modified OPRO (green). The results show that setting the temperature to 0 while selecting the last action has negligible impact on the performance of OPRO. Importantly, **our EXPO and EXPO-ES algorithms consistently and dramatically outperform all variants of OPRO**.

## E.3    IMPACT OF ADDING EXEMPLAR EMBEDDING TO THE NN IN EXPO

Recall that in every iteration of our EXPO (Algo. 1), we need to train a neural network (NN) $\mathcal{M}(g(\cdot); \theta)$ to estimate the scores of the task descriptions and meta-instructions in the domain (line 8 of Algo. 1). Note that the training set used to train this NN is $\{(\left[g(\mathcal{D}_i) \oplus g(\mathcal{I}_i)\right], s_i)\}_{i=1,\ldots,t+1}$ (line 7 of Algo. 1). However, it is also important to note that in our EXPO algorithm, the set of exemplars included in the meta-prompt $\mathcal{E}'_t$ changes in every iteration and hence may also affect the scores $s_i$'s. Therefore, one may naturally wonder whether including the embedding of $\mathcal{E}'_t$ in the input of the NN can further improve the performance of the trained NN and, consequently, the performance of the overall EXPO. We conduct an ablation study to validate this hypothesis, and the results are shown in Fig. 16. The results demonstrate that including the embedding of the exemplars in the input of the NN does not lead to better performance than our standard approach of excluding it (Algo. 1). This is likely due to the significantly increased dimensionality of the input to the NN, which makes training the NN more challenging. Therefore, these results suggest that the benefit of additionally accounting for the changing exemplars is outweighed by the drawback of the significantly increased dimensionality of the input to the NN.

## E.4    COMPUTATIONAL EFFICIENCY OF OUR ALGORITHMS

To compare the computational costs of our EXPO and OPRO, using the Linear Regression experiment ($w = 2, b = 30$), we recorded the time required for each method to achieve a mean squared error (MSE) below various thresholds. The results are shown in Table 1 below.

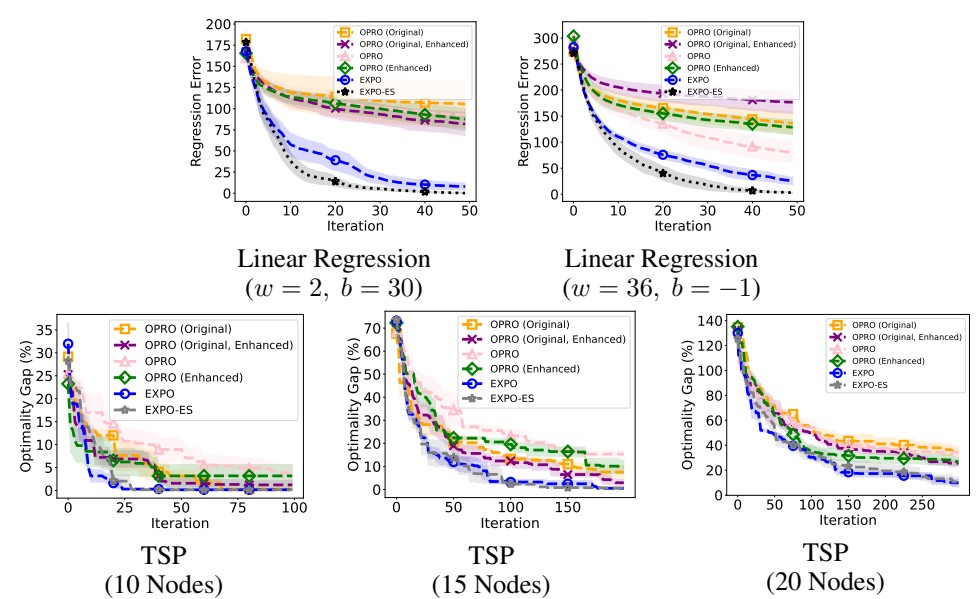

Figure 15: Results of different algorithms in the Linear Regression task and TSP task (Sec. 4.1). We have additionally included the original OPRO (which selects all 8 actions using a temperature of 1), as well as its enhanced variant. Lower is better.

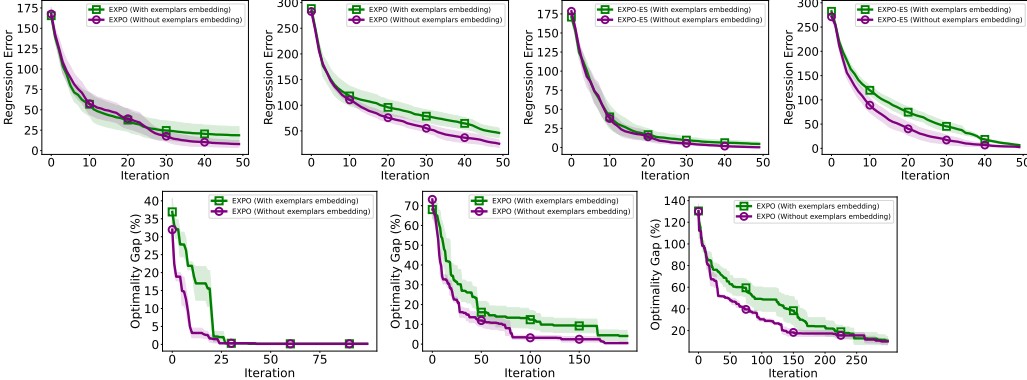

Figure 16: Convergence curves of our EXPO with and without exemplars embedding across different tasks: Linear Regression (top row) and TSP with 10, 15, and 20 nodes (bottom row).

The results show that EXPO requires only 263.1 seconds to achieve MSE $\leq 25$, whereas OPRO is unable to achieve this threshold even after 731.4 seconds. Similarly, EXPO achieves MSE $\leq 10$ in just 350.8 seconds, which is also less than OPRO's total runtime of 731.4 seconds (after 300 iterations). Therefore, to reach the same performance level, our EXPO requires significantly smaller amount of computational time than OPRO.

### E.5 MORE ILLUSTRATIONS OF THE DISCOVERED TASK DESCRIPTION AND META-INSTRUCTION

Here we provide more illustrations regarding the comparison of the original task description and meta-instruction adopted by the original LLM-based sequential decision-making algorithm (i.e., OPRO or the LLM-based MAB algorithm from Krishnamurthy et al. (2024)) and those optimized by our EXPO algorithm. We include the comparisons for the TSP task (Fig. 17), and the two different prompt designs for the LLM-based MAB task in Sec. 4.2 (Fig. 18 and Fig. 19).

| Methods | Time (s) to MSE $\leq$ 100 | Time (s) to MSE $\leq$ 50 | Time (s) to MSE $\leq$ 25 | Time (s) to MSE $\leq$ 10 | Total Runtime (s) |
|---|---|---|---|---|---|
| **OPRO** | 64.1 | 170.1 | NA | NA | 731.4 |
| **EXPO (Ours)** | 94.5 | 207.4 | 263.1 | 350.8 | 1645.6 |

Table 1: Comparison of our `EXPO` and OPRO in terms of the time (s) to reach different MSE thresholds and total runtime.

| OPRO | EXPO |
|---|---|
| You are given a list of points with coordinates below: {POINTS}.
Below are some previous traces and their lengths. The traces are arranged in descending order based on their lengths, where lower values are better.


{EXEMPLARS}

Give me a new trace that is different from all traces above, and has a length lower than any of the above. The trace should traverse all points exactly once. The trace should start with `<trace>` and end with `</trace>`. | You are provided with a dataset containing a list of coordinates labeled as {POINTS}.
The dataset also includes a series of previously calculated routes, with associated lengths that are ordered from longest to shortest. However, it's key to note that shorter routes are more desirable. Despite the presentation order, understand that the optimal route is identified by the smallest total length.

{EXEMPLARS}

Provide a unique trace that is distinct from any previous traces and shorter in length. Ensure that this trace visits each point exactly once and adhere to the specified format by starting with `<trace>` and concluding with `</trace>`. |

Figure 17: The task description (top) and meta-instruction (bottom) used by OPRO (left) and optimized by our `EXPO` (right) in a TSP task.

### E.6 MORE RESULTS ON THE ABLATION STUDY REGARDING COMPARISON WITH GENERAL PROMPT OPTIMIZATION METHODS

Here we use the TSP task to perform an additional ablation study to compare the performance of our `EXPO` algorithm with the INSTINCT algorithm (Lin et al., 2024b), which is the best-performing baseline method from Fig. 7. The results are shown in Fig. 20, which, together with Fig. 7, demonstrate that our `EXPO` algorithm based on adversarial bandits significantly and consistently outperforms general prompt optimization methods, including the INSTINCT algorithm which is based on stochastic MAB.

BSSND

You are a bandit algorithm in a room with 5 buttons labeled blue, green, red, yellow, purple. Each button is associated with a Bernoulli distribution with a fixed but unknown mean; the means for the buttons could be different. For each button, when you press it, you will get a reward that is sampled from the button's associated distribution. You have 100 time steps and, on each time step, you can choose any button and receive the reward. Your goal is to maximize the total reward over the 100 time steps.

At each time step, I will show you a summary of your past choices and rewards. Then you must make the next choice. You may output a distribution over the 5 buttons formatted EXACTLY like "blue:a,green:b,red:c,yellow:d,purple:e".

EXPO

You are presented as a bandit algorithm, located in an environment offering five distinct buttons, each emblazoned with colors such as blue, green, red, yellow, and purple. Each button acts a vessel tied to a non-variable yet undisclosed Bernoulli distribution mean which isn't subjected to be uniformly distributed across buttons. In this mechanism, every button acts as a yielder of a capricious reward, constructed from the associated distribution of the respective button. With access to total life -encompassing around 100 temporal stages - your voluntary element grants you control towards opting the button insertion at each such progressive phase. Precisely summoning your approach could perpetually provide you with a regulatory provision_, the aptitude - is flexibly dwelling within its underlining motive- aiming at optimizing total accumulated cashbacks during several phases of these 100 spatial temporalities.

During every step of the process, a recap highlighting your previous selections and the prizes received will be presented to you. Then, it'll now be incumbent upon you to proceed with the new decision-making process. For your ease, a well-structured distribution comprising five buttons in assorted colours such as "blue", "green,", "red", "yellow", and "purple" will be exhibited before you. Make sure to structure your output accordingly; this might look something akin to "blue:a,green:b,red:c,yellow:d,purple:e".

Figure 18: The suggestive framing (corresponding to the task description) and MAB problem description (corresponding to the meta-instruction) used by BSSND $hard$ (left) and optimized by our EXPO (right) in an LLM-based MAB task.

BSSCD

You are a bandit algorithm in a room with 5 buttons labeled blue, green, red, yellow, purple. Each button is associated with a Bernoulli distribution with a fixed but unknown mean; the means for the buttons could be different. For each button, when you press it, you will get a reward that is sampled from the button's associated distribution. You have 100 time steps and, on each time step, you can choose any button and receive the reward. Your goal is to maximize the total reward over the 100 time steps.

At each time step, I will show you a summary of your past choices and rewards. Then you must make the next choice. You may output a distribution over the 5 buttons formatted EXACTLY like "blue:a,green:b,red:c,yellow:d,purple:e". Let's think step by step to make sure we make a good choice.

EXPO

You are an algorithm designed to function as a bandit, positioned within an environment that features five distinct buttons, each colored blue, green, red, yellow, and purple. These buttons are intricately connected to individual Bernoulli distributions which possess unique and undisclosed mean probabilities. When a button is pressed, it delivers a reward based on its specific distribution. Granted with 100 opportunities to act, your objective is to strategically press these buttons in a manner that optimizes the accrued total reward throughout these attempts. Make your selections wisely to maximize the gains from this stochastic setup.

In each phase, a concise recap of your previous decisions and received rewards will be presented. Your task is to make a subsequent choice based on this data. It is essential to output your selection in an exact format defined as "blue:a, green:b, red:c, yellow:d, purple:e", where 'a', 'b', 'c', 'd', 'e' represent specific z-score values for each color accompanied by the decision choice letter(s). The process is designed to refine our strategy progressively with each move, ensuring an informed and impactful outcome.

Figure 19: The suggestive framing (corresponding to the task description) and MAB problem description (corresponding to the meta-instruction) used by BSSCD $hard$ (left) and optimized by our EXPO (right) in an LLM-based MAB task.

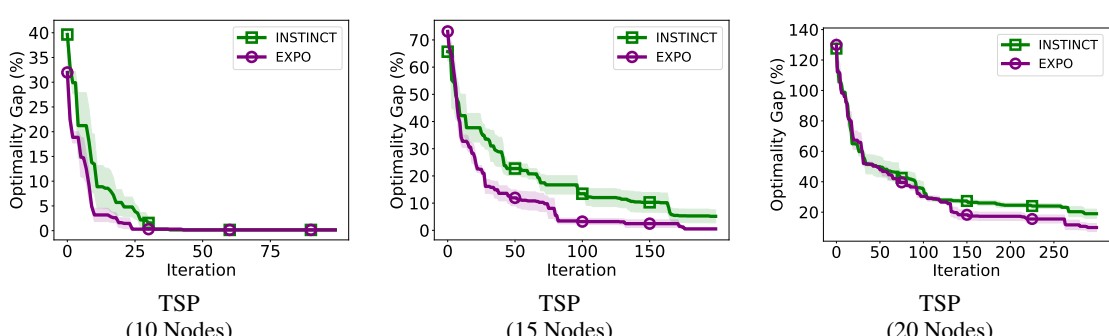

Figure 20: Comparison of our EXPO with INSTINCT in the TSP tasks.

