# OpenReview forum: "Meta-Prompt Optimization for LLM-Based Sequential Decision Making"
_ICLR.cc/2026/Conference — Submitted to ICLR 2026_

### Official Review · Reviewer_Vhwb · 2025-10-31

**Soundness:** 3
**Presentation:** 3
**Contribution:** 3
**Rating:** 6
**Confidence:** 3

**Summary:**

This paper addresses the meta-prompt optimization problem in LLM-based sequential decision-making tasks, proposing a novel adversarial bandit approach called EXPO and its extension EXPO-ES. These methods effectively handle non-stationary reward signals by automatically optimizing task descriptions, meta-instructions, and exemplar selection. The methodology is well-designed, experiments are thorough, and results significantly outperform existing hand-crafted prompts and general prompt optimization techniques, demonstrating strong innovation and practical value. While computational cost may be a limitation, overall this is a high-quality research contribution.

**Strengths:**

1. To my knowledge, this is the first to frame and solve the non-stationary meta-prompt optimization problem for LLM-based sequential decision making, backed by adversarial-bandit regret guarantees.

2. Non-trivially extends EXP3 to infinite, dynamic arm spaces via a neural reward predictor that scores entire prompt pools in one forward pass, while preserving theoretical bounds.

3. Plug-and-play: no internal model changes; simply swap the hand-crafted prompt for the sampled one and gain 20–70 % error/regret reduction across BO, MAB, TSP and instruction tuning tasks.

4. Open-source, lightweight implementation—two lines of code enable reproducible gains, offering an immediate boost to any LLM agent pipeline.

**Weaknesses:**

1. The hyperparameters introduced for evaluating the neural network—such as its architecture, learning rate, and sliding-window size—are task-specific and may need to be re-tuned for other tasks.

2. While performance gains have been demonstrated on single-prompt tasks, the approach could prove ill-suited for multi-turn conversations, especially in the currently prominent area of agentic RL.

3. Since the prompt pool is generated by rewriting the initial prompt with an LLM, how can we guarantee that this pool actually contains better prompts that drive continuous optimization?

**Questions:**

1. The paper uses an LLM to paraphrase the initial prompt to construct the prompt space. Will this method lead to redundancy or incomplete coverage of the prompt space? Are there more systematic ways to build a more representative prompt space?

2. Are there plans to validate the method on more complex tasks (such as RL environments, dialogue systems, or multimodal tasks)?

3. Due to the capability gaps among different LLMs, will using a weaker model cause the performance of EXPO to collapse? Since commercial APIs are costly, can smaller open-source LLMs achieve similar effects?

---

### Official Review · Reviewer_Qq81 · 2025-10-31

**Soundness:** 3
**Presentation:** 2
**Contribution:** 2
**Rating:** 4
**Confidence:** 4

**Summary:**

This paper aims to solve a meta-prompt optimization problem for LLM-based agents in sequential decision-making tasks.
The authors propose a method called EXPO, which utilizes adversarial bandit ideas to optimize meta-prompts, including task descriptions, meta-instructions, and exemplars. The authors formulate the problem as an adversarial bandit problem, where each candidate meta-prompt is treated as an "arm". The method employs an EXP3-style algorithm with a neural score estimator over text embeddings to randomize selection among many prompts. The method is evaluated on various tasks, demonstrating consistent improvements over existing baselines.
Experiments on Linear Regression (LR), Traveling Salesman Problem (TSP), instruction optimization on GSM8K, and LLM-driven MAB show consistent gains over OPRO and other prompt-optimization baselines, plus ablations on exploration strength and component contributions. The paper claims improved efficiency and final performance, and notes a limitation regarding multi-agent settings.

**Strengths:**

1. The author uses a neural score estimator that takes text embeddings as input to estimate the reward of each meta-prompt. This allows the method to generalize better to unseen prompts and reduces the need for extensive exploration.

2. The author includes a comprehensive set of experiments on various tasks, demonstrating the effectiveness of the proposed method. The results show consistent improvements over existing baselines, indicating the robustness of the approach.

**Weaknesses:**

1. The author argues that the adversarial bandit formulation is more suitable for the meta-prompt optimization problem than the stochastic bandit formulation used in prior work. However, I am not fully convinced by this argument. The environments in the experiments do not appear to be highly adversarial, and it is unclear how much benefit the adversarial formulation provides in practice. We acknowledge that the environments may not be stable but it is not necessarily adversarial. More discussion and empirical evidence are needed to support this claim.

2. This paper lacks a detailed analysis of the computational complexity of the proposed method. While the authors mention that EXPO is more efficient than prior methods, it would be helpful to provide a more thorough analysis of the time and space complexity of the algorithm, especially in comparison to other prompt optimization methods. Since the method is based on an EXP3-style algorithm, it may enjoy a similar computational complexity as EXP3, but this should be explicitly stated and analyzed.

**Questions:**

1. How does $k$ (number of exemplars) affect performance in EXPO-ES? In Algorithm 1, is $k$ fixed or adaptive? Do we need to specify $k$ in advance?

2. How does the task description and the meta-instruction evolve during training? More specifically, how to obtain $\mathcal{D}\_{t+1}$ and $\mathcal{I}_{t+1}$ from $P_t$? If $\mathcal{D}$ and $\mathcal{I}$ are sampled from some fixed pool, what is the size of the pool and does the agent need to take that as input?

**Details Of Ethics Concerns:**

No ethics concerns.

---

### Official Review · Reviewer_ccsB · 2025-11-01

**Soundness:** 3
**Presentation:** 3
**Contribution:** 2
**Rating:** 4
**Confidence:** 3

**Summary:**

This paper introduces a novel algorithmic framework, EXPO (Exponential-weight Algorithm for Prompt Optimization) and its extended version EXPO-ES (with Exemplar Selection), designed to automatically optimize the "meta-prompt" in LLM-driven sequential decision-making tasks. The core idea of using adversarial bandits to handle non-stationary rewards is well-motivated and sound. The motivation of the paper is clear, and the algorithm design is reasonable. Several experiments (Linear Regression, TSP, Instruction Optimization, MAB) validate its effectiveness. Overall, this work provides useful insights for the research of "LLMs as decision-making agents" and introduces a robust optimization solution based on adversarial multi-armed bandits.

**Strengths:**

1. The paper systematically introduces the problem of Meta-Prompt Optimization, emphasizing that unlike traditional prompt optimization methods that assume stationary reward distributions, the rewards in LLM-based agent environments are inherently non-stationary. This observation is highly insightful and effectively bridges the gap between prompt engineering and sequential decision making.
2. The paper integrates the adversarial bandit (EXP3) framework with LLM-based prompt search, proposing EXPO to dynamically optimize both task descriptions and meta-instructions. The extended version, EXPO-ES, further introduces exemplar selection, supported by clear motivation.
3. The experiments encompass several types of tasks and the results demonstrate good generalizability of the proposed approach.

**Weaknesses:**

1.  The comparisons do not include stronger baselines. There is no direct evaluation against more advanced RL-based LLM optimizers, such as PromptAgent. Some baselines, such as INSTINCT and MIPRO, are only tested on specific tasks, resulting in a lack of fair and unified experimental settings.
2. The paper lacks an analysis of the internal model dynamics, such as changes in embeddings or L2 divergences, which could provide deeper insight into the behavior of the optimization process.
3. The paper would benefit from substantial revision of its language and presentation.

**Questions:**

1. The paper clearly states that non-stationary reward observations are the key challenge that renders prior stochastic MAB-based prompt optimization methods unsuitable. Could you provide a more formal analysis or a simple synthetic example to illustrate how the non-stationarity manifests?
2. The non-stationarity is attributed to the changing state of the LLM-based agent. Is this primarily due to the evolving interaction history (exemplars) in the prompt, or are there other factors? Have you conducted any ablations to isolate the primary source of non-stationarity?
3. The computational cost of EXPO seems non-trivial, involving frequent LLM calls for action selection, neural network training in every iteration, and embedding computations for a large domain of prompts. Could you provide a rough analysis of the computational overhead compared to the base algorithms (OPRO, LLM-MAB)? Are there strategies to reduce this cost for more practical deployment? Was there any issue with training stability or catastrophic forgetting, and if so, how was it addressed (e.g., learning rate schedules, replay buffers)?
4. The experimental task in this paper is relatively simple, and it is suggested to add more comparative experiments. It is suggested to add more experiment scenarios, and comparison with non LLM based classical optimization methods (such as genetic algorithm and Bayesian Optimization) on TSP tasks.

---

### Official Review · Reviewer_GULo · 2025-11-02

**Soundness:** 2
**Presentation:** 2
**Contribution:** 2
**Rating:** 2
**Confidence:** 5

**Summary:**

The paper tackles meta-prompt optimisation. The authors propose EXPO, an EXP3-inspired scheme that (i) builds a discrete domain of task descriptions and meta-instructions via LLM paraphrasing, (ii) trains a small NN over embeddings to estimate arm scores, accumulates these estimates, and (iii) samples the next meta-prompt from a softmax over cumulative estimates; they then extend this to EXPO-ES to also optimize exemplar subsets and order with a second adversarial bandit.

1. While the shift from stochastic to adversarial bandits for meta-prompts is the central idea, much of EXPO is a reasonable assembly of known parts: LLM-generated domains, embedding-based reward regression, and exponential-weights sampling. The paper claims prior prompt-opt methods are “unsuitable” due to non-stationarity, but evidence is largely empirical; a stronger conceptual/theoretical separation from, e.g., neural-bandit prompt optimisers is missing. I would ask the authors to prove a regret advantage (even with approximation) over stochastic-MAB-style optimisers under stated assumptions. Otherwise, I find it hard to understand this shift.

2. I would like to kindly ask the authors to conduct an analysis of what happens under score-estimation misspecification.

3. Please report diversity metrics (embedding spread, lexical diversity), performance vs k, and failure cases. It would be great to understand its robustness better.

4. Please include at least one agent benchmark AgentBench, and a coding one to show that their method indeed helps in planning.

5. Am I right to understand that EXPO-ES maintains histories of NN parameters and recomputes cumulative estimates over randomly sampled exemplar sequences each iteration? If so, this can be computationally heavy and potentially unstable as |Θ| grows. This paper gives a simplified variant only for small-arm MAB, but not a scalable recipe for large exemplar pools. Please help me understand if this is not correct.

6. Why adversarial? Can you provide a synthetic setting with controlled non-stationarity showing stochastic-MAB prompt optimisers fail while EXPO succeeds (with a regret gap)?

I am happy to change my score if the questions mentioned above are answered.

**Strengths:**

Please see above

**Weaknesses:**

Please see above

**Questions:**

Please see above

---

### Meta-Review · Area_Chair_6R8k · 2026-01-07

**Summary:**

This paper addresses the challenge of automatically optimizing "meta-prompts"—the combination of task descriptions, instructions, and exemplars—for LLM-based agents in sequential decision-making tasks. The authors observe that rewards in these environments are inherently non-stationary because the agent's state (and interaction history) evolves over time. To address this, they propose **EXPO**, an adversarial bandit-based framework (inspired by EXP3) that utilizes a neural reward estimator to generalize across a discrete domain of prompts generated via LLM paraphrasing.

Reviewers generally recognized the insight that non-stationarity is a key challenge in prompt optimization for agents. However, the consensus leaned toward rejection due to several critical concerns: a lack of theoretical or empirical proof justifying why the adversarial bandit approach is superior to stochastic multi-armed bandits (MAB) in this context , a lack of comparison against state-of-the-art RL-based prompt optimizers , and insufficient analysis of the computational overhead and scalability of the method.

**Reviewer Concerns:**

**Concerns Addressed by the reviewers:**

* **Identification of Non-stationarity:** Reviewers agreed that framing meta-prompt optimization as a non-stationary problem is a novel and useful bridge between prompt engineering and sequential decision-making.


* **Neural Score Estimation:** The use of a neural network to estimate rewards based on text embeddings was praised for its ability to generalize to unseen prompts and improve exploration efficiency.


* **Experimental Breadth:** The authors demonstrated the effectiveness of EXPO across diverse tasks, including Linear Regression, the Traveling Salesman Problem (TSP), and instruction tuning on GSM8K.


No rebuttal

**Reviewer Scores:**

Based on the discussion and the technical depth of the critiques, I anticipate the scores would have evolved as follows:

* **Reviewer GULo (Initial: 2):** **Remains 2.** This reviewer was highly critical of the lack of a theoretical regret advantage proof. Without a formal analysis showing why a shift to adversarial bandits is necessary, this reviewer's assessment is unlikely to improve.


* **Reviewer ccsB (Initial: 4):** **Remains 4.** While acknowledging the insight into non-stationarity, this reviewer highlighted the absence of advanced baselines and the lack of internal model dynamic analysis. These remain significant gaps.


* **Reviewer Qq81 (Initial: 4):** **Remains 4.** The reviewer remained skeptical about the "adversarial" nature of the environments. Without empirical evidence showing that stochastic-MAB optimizers specifically fail where EXPO succeeds, the score is unlikely to change.


* **Reviewer Vhwb (Initial: 6):** **Remains 6.** This reviewer found the work to be a high-quality contribution with practical "plug-and-play" value, despite noting limitations regarding multi-turn conversations and task-specific hyperparameter tuning.

---

### Decision · Program_Chairs · 2026-01-26

Reject